# NLRP12 suppresses hepatocellular carcinoma via downregulation of cJun N-terminal kinase activation in the hepatocyte

SM Nashir Udden[1], Youn-Tae Kwak[1,2], Victoria Godfrey[1], Md Abdul Wadud Khan[3], Shahanshah Khan[1], Nicolas Loof[4], Lan Peng[1], Hao Zhu[4,5,6,7], Hasan Zaki[1]*

[1]Department of Pathology, UT Southwestern Medical Center, Dallas, United States; [2]Department of Biochemistry, UT Southwestern Medical Center, Dallas, United States; [3]Department of Surgical Oncology, MD Anderson Cancer Center, Houston, United States; [4]Children's Research Institute, UT Southwestern Medical Center, Dallas, United States; [5]Department of Pediatrics, UT Southwestern Medical Center, Dallas, United States; [6]Center for Regenerative Science and Medicine, UT Southwestern Medical Center, Dallas, United States; [7]Department of Internal Medicine, UT Southwestern Medical Center, Dallas, United States

**Abstract** Hepatocellular carcinoma (HCC) is a deadly human cancer associated with chronic inflammation. The cytosolic pathogen sensor NLRP12 has emerged as a negative regulator of inflammation, but its role in HCC is unknown. Here we investigated the role of NLRP12 in HCC using mouse models of HCC induced by carcinogen diethylnitrosamine (DEN). $Nlrp12^{-/-}$ mice were highly susceptible to DEN-induced HCC with increased inflammation, hepatocyte proliferation, and tumor burden. Consistently, $Nlrp12^{-/-}$ tumors showed higher expression of proto-oncogenes cJun and cMyc and downregulation of tumor suppressor p21. Interestingly, antibiotics treatment dramatically diminished tumorigenesis in $Nlrp12^{-/-}$ mouse livers. Signaling analyses demonstrated higher JNK activation in $Nlrp12^{-/-}$ HCC and cultured hepatocytes during stimulation with microbial pattern molecules. JNK inhibition or NLRP12 overexpression reduced proliferative and inflammatory responses of $Nlrp12^{-/-}$ hepatocytes. In summary, NLRP12 negatively regulates HCC pathogenesis via downregulation of JNK-dependent inflammation and proliferation of hepatocytes.
DOI: https://doi.org/10.7554/eLife.40396.001

*For correspondence:
hasan.zaki@utsouthwestern.edu

Competing interests: The authors declare that no competing interests exist.

## Introduction

Hepatocellular carcinoma (HCC) is the 5th most common malignancy and the 3rd most common cause of cancer-related death worldwide (*Farazi and DePinho, 2006*). Major risk factors for HCC include hepatitis B and C virus (HBV and HCV) infections, obesity, alcohol abuse, and drug toxicity (*El-Serag and Rudolph, 2007*; *Farazi and DePinho, 2006*). The common physiological process prior to HCC development is inflammation, which triggers DNA damage and mutagenesis, hepatic cell death, and compensatory proliferation (*Maeda et al., 2005*; *Sakurai et al., 2008*). Consistently, inflammatory signaling pathways including NF-κB, MAPK, STAT3 and AKT are hyperactivated in HCC and considered critical players in HCC pathogenesis (*Calvisi et al., 2006*; *He and Karin, 2011*; *Hui et al., 2008*; *Maeda et al., 2005*; *Pikarsky et al., 2004*; *Wang et al., 2016*).

A major class of stimuli for these pathways are pathogen-associated molecular patterns (PAMPs) which are sensed by pattern recognition receptors (PRRs). The best-characterized PRRs are toll-like

receptors (TLRs) that sense a wide array of PAMPs on the cell surface or endosomal compartment (*Akira et al., 2006*). Because of its close anatomical connection with the intestine, the liver is constantly exposed to gut microbiota-derived PAMPs (*Son et al., 2010*). The translocation of microbes and their PAMPs is enhanced during chronic liver disorders (*Campillo et al., 1999*; *Cirera et al., 2001*; *Fukui et al., 1991*; *Pascual et al., 2003*; *Rutenburg et al., 1957*; *Yoneyama et al., 2002*) and likely an important contributor of HCC development. Indeed, sensing lipopolysaccharide (LPS) by liver parenchymal cell-specific TLR4 contributes to liver fibrosis and HCC (*Dapito et al., 2012*; *Machida et al., 2009*; *Paik et al., 2003*; *Seki et al., 2007*). However, the regulatory mechanisms in the gut-liver inflammatory and carcinogenic axis are poorly explored but critical to understanding and potentially treating HCC.

In addition to TLRs, several cytosolic PRRs sense and respond to PAMPs or danger-associated molecular patterns (DAMPs) and activate downstream cell signaling pathways. The NOD-like receptors (NLRs) are a family of cytosolic PRRs which are associated with diverse diseases related to infections, inflammation and cancer (*Saxena and Yeretssian, 2014*; *Zhong et al., 2013*). The NLR family member NLRP12 has recently emerged as a critical regulator of inflammation and cancer. Previous studies showed that deficiency of NLRP12 leads to enhanced incidence and faster progression of colorectal tumorigenesis (*Allen et al., 2012*; *Chen et al., 2017*; *Zaki et al., 2011*). NLRP12 negatively regulates NF-κB and ERK in macrophages, dendritic cells, and T cells (*Lukens et al., 2015*; *Zaki et al., 2014*; *Zaki et al., 2011*), and increased colorectal tumorigenesis in $Nlrp12^{-/-}$ mice is associated with higher activation of the NF-κB and ERK signaling pathways (*Allen et al., 2012*; *Zaki et al., 2011*). In the liver, NLRP12 is highly expressed and dampens inflammatory responses secondary to *Salmonella* Typhimurium infection (*Zaki et al., 2014*). These observations suggest that NLRP12 may regulate inflammatory disorders of the liver such as HCC.

Here, we investigated the role of NLRP12 in HCC using mouse models. The expression of NLRP12 was seen negatively correlated with human and mouse HCC. $Nlrp12^{-/-}$ mice developed significantly higher tumor burden in the liver following administration of mutagens. HCC susceptibility in $Nlrp12^{-/-}$ mice was eliminated with antibiotics treatment. Our in vivo and in vitro data demonstrated that NLRP12 suppresses PAMP-mediated proliferation and inflammatory gene expression in hepatocytes via attenuation of JNK signaling. This study underscores a novel cancer suppressive pathway in the liver involving NLRP12.

## Results

### The loss of NLRP12 is associated with increased HCC susceptibility

To understand an association of NLRP12 with human HCC, we analyzed publicly available cancer genomics databases. According to The Cancer Genome Atlas (TCGA) database, about 2% of HCC patients carry mutations in *NLRP12* (*Figure 1A*). Analysis of RNA-seq data in the TCGA database using the UALCAN web-portal (*Chandrashekar et al., 2017*) revealed that the expression of NLRP12 is significantly (p=0.0004) reduced in human HCC (*Figure 1B*). To mechanistically characterize the role of NLRP12 in HCC, we used a mouse model in which HCC was induced with the administration of a single dose of diethylnitrosamine (DEN) (*Figure 1—figure supplement 1A*). DEN is a procarcinogen that induces DNA damage and cell death in the liver, leading to the development of HCC (*Bakiri and Wagner, 2013*; *Rajewsky et al., 1966*). 10 months post a single DEN injection into WT and $Nlrp12^{-/-}$ mice, we collected whole livers and measured the number and size of tumors. Consistent to reduced *NLRP12* in human HCC, the expression of *Nlrp12* was significantly reduced in DEN-induced HCC compared to healthy livers of WT mice (*Figure 1C*). As we counted the number of visible tumors, we observed significantly higher number of tumors in $Nlrp12^{-/-}$ mouse livers compared to that of WT mice (*Figure 1D and E*). Tumor sizes and tumor/body weight ratios of $Nlrp12^{-/-}$ mice were significantly larger compared to those of WT mice (*Figure 1E*). The areas of adenoma in $Nlrp12^{-/-}$ livers were significantly larger than that of WT (*Figure 1F and G*). HCC is associated with liver damage leading to the elevation of serum levels of ALT and AST. As expected, ALT and AST levels were significantly higher in $Nlrp12^{-/-}$ mice at 10 months after DEN administration (*Figure 1H*). We confirmed the role of NLRP12 in HCC development in a second model involving carbon tetrachloride ($CCl_4$) along with DEN (*Figure 1—figure supplement 1B*). $CCl_4$ is a toxic chemical which causes hepatic necrosis, compensatory hepatocyte proliferation, and ultimately drives fibrosis

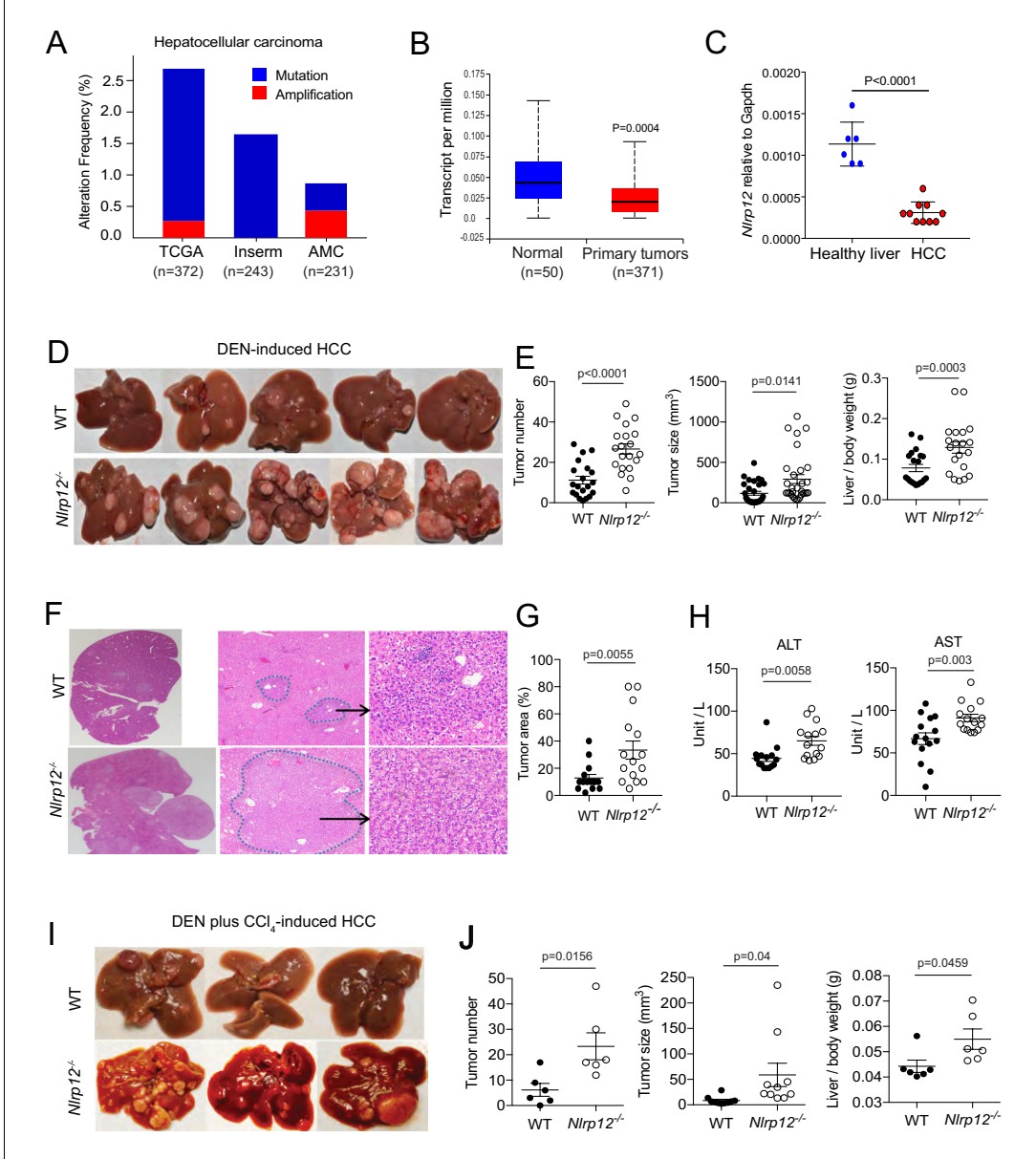

**Figure 1.** NLRP12 negatively regulates hepatocellular carcinoma. (**A**) Analysis of genetic association of *NLRP12* in HCC using the TCGA database through the cBioportal online platform. (**B**) Analysis of *NLRP12* expression in human normal liver and HCC RNA-seq data in the TCGA database through the UALCAN web-portal. Data are presented in a box plot where whiskers represent maximum and minimum variables. (**C**) WT mice were injected with DEN (25 mg/kg i.p.) at the age of 14 days or left untreated. Livers from DEN-treated (n = 10) and untreated mice (n = 6) were analyzed for the expression of Nlrp12 by real-time qPCR. Data represent means ± SEM. (**D–H**) WT mice (n = 20) and *Nlrp12*-/- (n = 20) were injected with DEN (25 mg/kg i.p.) at the age of 14 days and sacrificed at 10 months after DEN injection. (**D**) Representative images of liver tumors are shown. (**E**) The number of tumors per liver, tumor sizes, and liver to body weight ratios were measured. Data represent means ± SEM (n = 20). Statistical difference was determined by two-tailed unpaired t-test. (**F**) Livers from mice described in D were stained for H and E. Representative H and E-stained sections are shown. (**G**) H and E-stained liver sections were histopathologically examined for adenoma development. Data represent means ± SEM (n = 15). Statistical difference was determined by two-tailed unpaired t-test. (**H**) Serum ALT and AST levels in tumor-bearing mice. Data represent means ± SEM (n = 15). Statistical difference was determined by two-tailed unpaired t-test. (**I–J**) WT (n = 6) and *Nlrp12*-/- (n = 6) mice were injected with DEN (25 mg/kg i.p.) at the age of 14 days followed by eight weekly injections of $CCl_4$ (0.5 ml/kg i.p., dissolved in corn oil) starting at 10 weeks of age and euthanized at the age of 6 months. (**I**) Representative images of liver tumors are shown. (**J**) The numbers, sizes, and liver to body weight ratio were measured. Data represent means ± SEM. Statistical difference was determined by two-tailed unpaired t-test.

DOI: https://doi.org/10.7554/eLife.40396.002

The following source data and figure supplements are available for figure 1:

*Figure 1 continued on next page*

*Figure 1 continued*

**Source data 1.** NLRP12 negatively regulates hepatocellular carcinoma.
DOI: https://doi.org/10.7554/eLife.40396.005
**Figure supplement 1.** Nlrp12-deficiency doesn't induce HCC in healthy untreated control mice.
DOI: https://doi.org/10.7554/eLife.40396.003
**Figure supplement 1—source data 1.** Nlrp12-deficiency doesn't alter liver fucntion in healthy mice.
DOI: https://doi.org/10.7554/eLife.40396.004

(*Sarma et al., 1986*). Similar to that seen in the DEN model, DEN plus CCl$_4$-treated *Nlrp12$^{-/-}$* mice developed a greater tumor burden with significantly larger tumors than WT mice (*Figure 1I and J*). Notably, control (DEN-untreated) *Nlrp12$^{-/-}$* mice did not develop any tumors and did not exhibit elevated ALT and AST levels (*Figure 1—figure supplement 1C–E*). These results suggest that NLRP12 plays a protective role against carcinogen-induced HCC in mice.

## Increased tumorigenesis in *Nlrp12$^{-/-}$* mice is associated with higher inflammation

Development of HCC is a multistep process involving hepatic steatosis, fibrosis, and cirrhosis (*Capece et al., 2013*; *El-Serag and Rudolph, 2007*). To characterize the role of NLRP12 in HCC histopathologically, we examined the H and E-stained liver sections of WT and *Nlrp12$^{-/-}$* mice collected at 10 months after DEN injection. There were significantly higher inflammatory infiltrates, steatosis, and fibrosis in *Nlrp12$^{-/-}$* livers relative to WT (*Figure 2A and B*). Similarly, DEN plus CCl$_4$-treated *Nlrp12$^{-/-}$* livers exhibited worsened histopathology (*Figure 2—figure supplement 1A–C*). Proinflammatory mediators such as IL-1$\alpha$, IL-6, TNF$\alpha$, and Cox2 have been implicated in HCC (*Bromberg and Wang, 2009*; *He et al., 2013*; *Luedde and Schwabe, 2011*; *Park et al., 2010*; *Sakurai et al., 2008*). To understand the nature of inflammatory responses in the context of NLRP12-deficiency, we measured different cytokines, chemokines, and inflammatory mediators in the tumor-bearing livers from DEN or DEN plus CCl$_4$-treated mice. Consistent with pathological features, there was significantly higher expression of cytokines IL-6 (*Il6*) and TNF$\alpha$ (*Tnfa*), chemokines KC (*Cxcl1*), MIP2 (*Cxcl2*), and MCP1 (*Ccl2*), and tumor-promoting molecule COX2 (*Cox2*) in *Nlrp12$^{-/-}$* HCC (*Figure 2C* and *Figure 2—figure supplement 1D*). Higher protein levels of IL-6, TNF$\alpha$, and KC in *Nlrp12$^{-/-}$* HCC tissue were confirmed by ELISA (*Figure 2—figure supplement 1E*). However, no differences were observed in the levels of T cell-dependent cytokines IL-4, IFN$\gamma$, and IL-17 (*Figure 2—figure supplement 1F*).

We next characterized different immune cell populations infiltrated into tumor tissues by flow cytometry. No differences were found in the number of neutrophils (Gr1-positive cells) and T-cells (TCR$\beta$-positive) (*Figure 2D and E*). However, there were more F4/80-positive macrophages (Kupffer cells) and CD11c-positive dendritic cells in *Nlrp12$^{-/-}$* HCCs (*Figure 2D and E*). Immunostaining and real-time qPCR analysis for F4/80 confirmed increased number of Kupffer cells in *Nlrp12$^{-/-}$* HCC as compared to WT (*Figure 2F and G*). F4/80 (*Emr1*) mRNA levels were unchanged between healthy WT and *Nlrp12$^{-/-}$* livers (*Figure 2—figure supplement 1G*), suggesting that increased expression of macrophage chemoattractant proteins MIP2 and MCP1 in *Nlrp12$^{-/-}$* HCC tissue resulted in increased infiltration of macrophages/Kupffer cells. These data indicate that NLRP12-deficiency promotes inflammatory responses during DEN-induced liver injury and tumorigenesis.

## NLRP12 regulates proliferation and cell death in the liver

DEN-induced liver injury is thought to promote hepatocarcinogenesis by inducing compensatory proliferation after widespread apoptosis (*Maeda et al., 2005*; *Sakurai et al., 2008*). To understand whether sensing PAMPs by NLRP12 regulates proliferative responses during HCC, we immunostained healthy and HCC livers for Ki67. The number of Ki67-positive cells were significantly higher in livers containing HCC compared to healthy livers of either WT or *Nlrp12$^{-/-}$* mice (*Figure 3A*). However, NLRP12-deficiency significantly increased hepatocyte proliferation in tumor tissues (*Figure 3A*). Higher expression of Ki67 in *Nlrp12$^{-/-}$* tumors was corroborated by real-time qPCR (*Figure 3B*). We further assessed the role of NLRP12 in proliferation by BrdU incorporation assay.

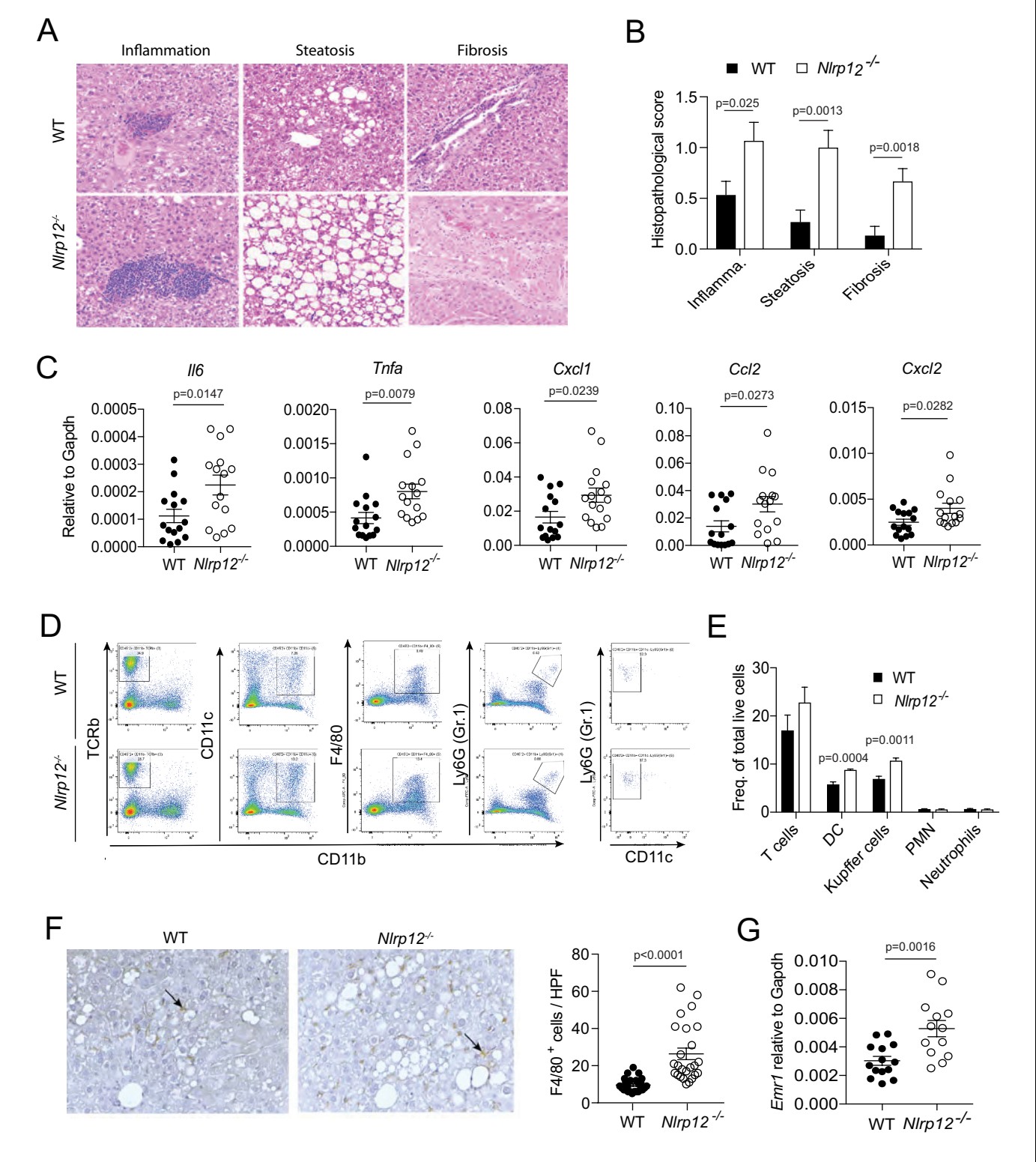

**Figure 2.** NLRP12-deficiency leads to increased inflammation in the liver. WT and *Nlrp12⁻/⁻* mice were injected with DEN (25 mg/kg i.p.) at the age of 14 days and euthanized 10 months later. (A–B) H and E-stained liver sections were examined histopathologically and scored for inflammation, steatosis, and fibrosis. Representative images of inflammation, steatosis, and fibrosis are shown. Data represent means ± SEM (n = 15). Statistical difference was determined by two-tailed unpaired t-test. (C) Real-time qPCR analysis of cytokines and chemokines in the HCC tissues. Data represent means ± SEM (n = 15). Statistical difference was determined by two-tailed unpaired t-test. (D) Tumor tissues were analyzed for the quantification of T cells (CD45⁺,

*Figure 2 continued on next page*

Figure 2 continued

TCRb[+], CD11b[-]), dendritic cells (CD45[+], CD11b[+], CD11c[+]high, Gr1[-]), Kupffer cells (CD45[+], CD11b[+], F4/80[+], Gr1[-]), and neutrophils (CD45[+], CD11b[+], Gr1[+], CD11c[-]) by flow cytometry. (E) Relative abundance of different immune cell types was analyzed by FlowJo software. Data represent means ± SEM (n = 5). Statistical difference was determined by two-tailed unpaired t-test. (F) Formalin-fixed and paraffin-embedded HCC sections were stained with F4/80 antibody and the number of F4/80[+] cells (brown) per high power field (20X) was counted. The picture shows representative immunostaining of F4/80 (brown). Data represent means ± SEM (n = 25). Statistical difference was determined by two-tailed unpaired t-test. (G) The expression of Emr1 (F4/80) in the HCC tissues was measured by real-time PCR. Data represent means ± SEM (n = 13–14). Statistical difference was determined by two-tailed unpaired t-test.

DOI: https://doi.org/10.7554/eLife.40396.006

The following source data and figure supplements are available for figure 2:

**Source data 1.** NLRP12-deficiency leads to increased inflammation in the liver.

DOI: https://doi.org/10.7554/eLife.40396.009

**Figure supplement 1.** NLRP12 suppresses inflammatory responses in HCC tissue.

DOI: https://doi.org/10.7554/eLife.40396.007

**Figure supplement 1—source data 1.** NLRP12 suppresses inflammatory responses during HCC.

DOI: https://doi.org/10.7554/eLife.40396.008

Similar to Ki67 staining, there was significantly higher BrdU-positive cells in *Nlrp12[-/-]* HCC tissues (*Figure 3—figure supplement 1A*).

To determine whether higher proliferation in *Nlrp12[-/-]* HCC is linked to cell death, we measured apoptosis by TUNEL assay in HCC livers collected at 10 months after DEN treatment. The number of TUNEL-positive cells in *Nlrp12[-/-]* HCC livers was significantly higher than WT, although no such difference was observed in healthy untreated WT and *Nlrp12[-/-]* mouse livers (*Figure 3C*). Increased cell death in *Nlrp12[-/-]* HCC was further demonstrated by immunohistochemical detection of cleaved cap-sase-3; *Nlrp12[-/-]* HCC livers had a significantly higher number of caspase-3-positive cells compared to livers from DEN-treated WT mice (*Figure 3D*). Furthermore, Western blot analysis showed higher caspase-3 cleavage and cytochrome C release in *Nlrp12[-/-]* HCC (*Figure 3E*).

Inflammatory mediators are known to contribute to hepatocyte cell death (*Kamata et al., 2005*; *Sakurai et al., 2008*). We wondered whether inflammation could promote apoptosis in *Nlrp12[-/-]* hepatocytes in a cell extrinsic fashion. Hence, we investigated the role of NLRP12 in hepatocyte cell death by culturing primary hepatocytes from WT and *Nlrp12[-/-]* mice and treating them with LPS. While unstimulated WT and *Nlrp12[-/-]* hepatocytes showed a similar level of TUNEL-positive cells, there was significantly higher apoptosis in LPS-treated *Nlrp12[-/-]* hepatocytes, suggesting that higher inflammatory responses in *Nlrp12[-/-]* hepatocytes during LPS stimulation contributed to increased cell death (*Figure 3—figure supplement 1B and C*). Altogether, these data imply that NLRP12 does not directly interfere with apoptosis, but indirectly modulates hepatic cell death during DEN-induced HCC via regulation of inflammatory responses.

## NLRP12 dampens oncogenic signals in the liver

Next, we focused on elucidating the molecular mechanisms of increased proliferation in *Nlrp12[-/-]* HCC. Alpha-fetoprotein (AFP), a marker of HCC, was significantly higher in *Nlrp12[-/-]* tumor tissue (*Figure 4A*). We measured the expression of proliferation-associated genes in WT and *Nlrp12[-/-]* HCC by real-time qPCR. There was increased expression of pro-proliferative and tumor promoting genes such as *Ccnb1, Ccnd1, Survivin,* and *Myc* (*Figure 4A*), while the expression of *Cdkn1a*, an inhibitor of cell cycle progression, was significantly reduced in *Nlrp12[-/-]* HCC (*Figure 4A*). The protein level of Ccnd1 (Cyclin d1) was also significantly higher in *Nlrp12[-/-]* tumors (*Figure 4B and C*). The expression of Cyclin d1 is regulated by several transcription factors including cMyc and cJun, which are major HCC relevant oncogenes (*Dang, 1999*; *Eferl et al., 2003*; *Lin et al., 2010*; *Schwabe et al., 2003*). Consistently, there was higher protein level of cMyc and activated cJun (P-cJun) in *Nlrp12[-/-]* HCC (*Figure 4B and C*).

Multiple pathways including NF-κB, ERK, JNK, p38, and STAT3 regulate the expression of proliferative genes and transcription factors involved in cancer (*Grivennikov et al., 2010*; *He and Karin, 2011*). To explore the pathways through which NLRP12 regulates oncogene expression, we measured the activation of NF-κB, ERK, p38, JNK, and STAT3, in HCC tissues from WT and *Nlrp12[-/-]* mice by Western blotting and ELISA. While all these pathways were activated in both WT and

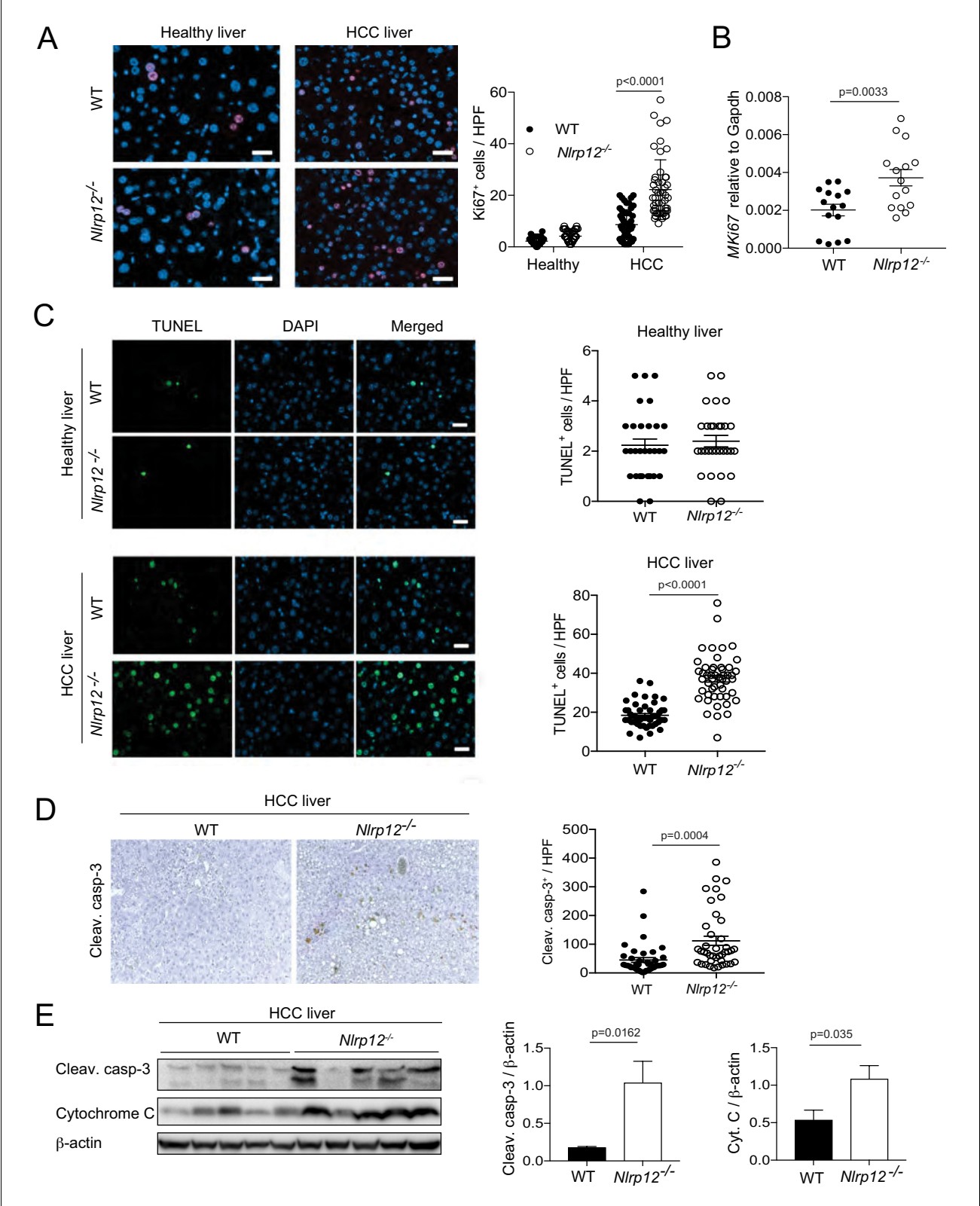

**Figure 3.** Increased HCC in Nlrp12-/- mice is associated with increased cell death and proliferation in the livers. WT and *Nlrp12*[-/-] were injected with DEN (25 mg/kg i.p.) or PBS (healthy control) at the age of 14 days and sacrificed at 10 months after DEN administration. (**A**) Liver tissue sections from healthy controls and DEN-treated mice were immunostained with Ki67 antibody and the number of Ki67-positive cells was counted under 20X objective. Data were collected from at least 10 fields per liver section and three mice/group. Data represent means ± SEM (n = 50). Statistical difference

*Figure 3 continued on next page*

*Figure 3 continued*

was determined by two-tailed unpaired t-test. (**B**) The expression of Ki67 in tumor tissues was measured by real-time qPCR. Data represent means ± SEM (n = 15; each sample represents individual mouse). Statistical difference was determined by two-tailed unpaired t-test. (**C**) Apoptosis in the healthy and HCC livers were measured by TUNEL assay. The number of TUNEL-positive cells (green) under 20X objective was counted and plotted as individual values. Data were collected from at least 10 fields per liver section and three mice/group. Data represent means ± SEM. Statistical difference was determined by two-tailed unpaired t-test. (**D**) Liver sections from DEN-treated mice (n = 3) were immunostained for cleaved caspase-3 (brown). Cleaved caspase-3 positive cells were counted under 20X objective. Data represent means ± SEM (n = 40). Statistical difference was determined by two-tailed unpaired t-test. (**E**) Liver tumor lysates were immunoblotted with anti-cleaved caspase-3, cytochrome c, and β-actin. The band intensities of caspase-3 and cytochrome c were measured. Data represent means ± SEM (n = 5; each sample represents individual mouse). Statistical difference was determined by two-tailed unpaired t-test.

DOI: https://doi.org/10.7554/eLife.40396.010

The following source data and figure supplements are available for figure 3:

**Source data 1.** Measurement of cell death and proliferation in WT and Nlrp12-deficient HCC.
DOI: https://doi.org/10.7554/eLife.40396.013
**Figure supplement 1.** NLRP12 regulates hepatocyte death and proliferation during HCC.
DOI: https://doi.org/10.7554/eLife.40396.011
**Figure supplement 1—source data 1.** NLRP12 regulates hepatocyte death and proliferation.
DOI: https://doi.org/10.7554/eLife.40396.012

*Nlrp12$^{-/-}$* HCC, the JNK pathway was consistently highly activated in *Nlrp12$^{-/-}$* tumors as compared to those of WT (*Figure 4D–F*). JNK plays critical roles in hepatocyte physiology by regulating cell death and proliferation, and deletion of JNK1 was shown to suppress HCC (*Hui et al., 2008*; *Sakurai et al., 2006*; *Schwabe, 2006*). Thus, the JNK pathway may be involved in NLRP12-mediated regulation of oncogene expression.

We hypothesized that JNK activation in *Nlrp12$^{-/-}$* HCC occurs in parenchymal tumor cells. To test this hypothesis, we isolated parenchymal cells from tumor tissue and measured the activation of JNK and other signaling pathways by Western blotting. *Nlrp12$^{-/-}$* tumor cells exhibited significantly increased phosphorylation of JNK, while no remarkable difference in the activation of ERK, p38, NF-κB pathways was observed (*Figure 4G and H*). Along with higher JNK activation, there was significantly increased expression of inflammatory cytokines *Cxcl1 and Ccl2*, and pro-proliferative molecules including *Survivin, Myc, Ccnd1, and MKi67* and reduced expression of *Cdkn1a* (*Figure 4I*). These levels coincide with the expression profiles of these molecules in the liver tumors (*Figures 2C* and *4A*).

To examine the role of NLRP12 in the regulation of JNK in non-parenchymal cells, such as Kupffer cells and hepatic stellate cells, in the tumor microenvironment, we isolated hepatocytes, Kupffer cells, and hepatic stellate cells from WT and *Nlrp12$^{-/-}$* HCC tissues. The purity of hepatocytes, Kupffer cells and hepatic stellate cells was confirmed by real-time qPCR of *Alb, Emr1, and Pdgfr-b* respectively (*Figure 4—figure supplement 1A*). NLRP12 was seen expressed in all these cell types (*Figure 4—figure supplement 1B*). Cells were then stimulated with LPS and activation of JNK was measured by Western blotting. Interestingly, higher JNK activation was seen only in hepatocytes but not in Kupffer cells and hepatic stellate cells (*Figure 4J*). LPS stimulation of tumor hepatocytes further enhanced the expression of inflammatory cytokines and chemokines *Ccl2, Cxcl1, Cxcl2, Tnfa, and Il6* (*Figure 4—figure supplement 1C*). Taken together, these findings suggest that higher activation of JNK in *Nlrp12$^{-/-}$* HCC contributes to increased expression of cytokines and chemokines, which help recruitment of macrophages and dendritic cells in the tumor microenvironment (*Figure 2E*), and tumor proliferation (*Figure 3A*).

## NLRP12 attenuates PAMPs-mediated hepatic inflammation and oncogenesis

HCC develops in a microenvironment of chronic liver injury, inflammation and fibrosis. However, existing evidence points to the critical contribution of PAMPs derived from gut microbiota in promoting HCC pathogenesis (*Tandon and Garcia-Tsao, 2008*). Sensing gut-derived LPS by TLR4 in the liver promotes inflammation, fibrosis and carcinogenesis (*Dapito et al., 2012*; *Fukui et al., 1991*; *Paik et al., 2003*; *Rutenburg et al., 1957*; *Seki et al., 2007*). Our data also suggest that LPS triggers inflammatory responses in tumor parenchymal cells (*Figure 4J*; *Figure 4—figure*

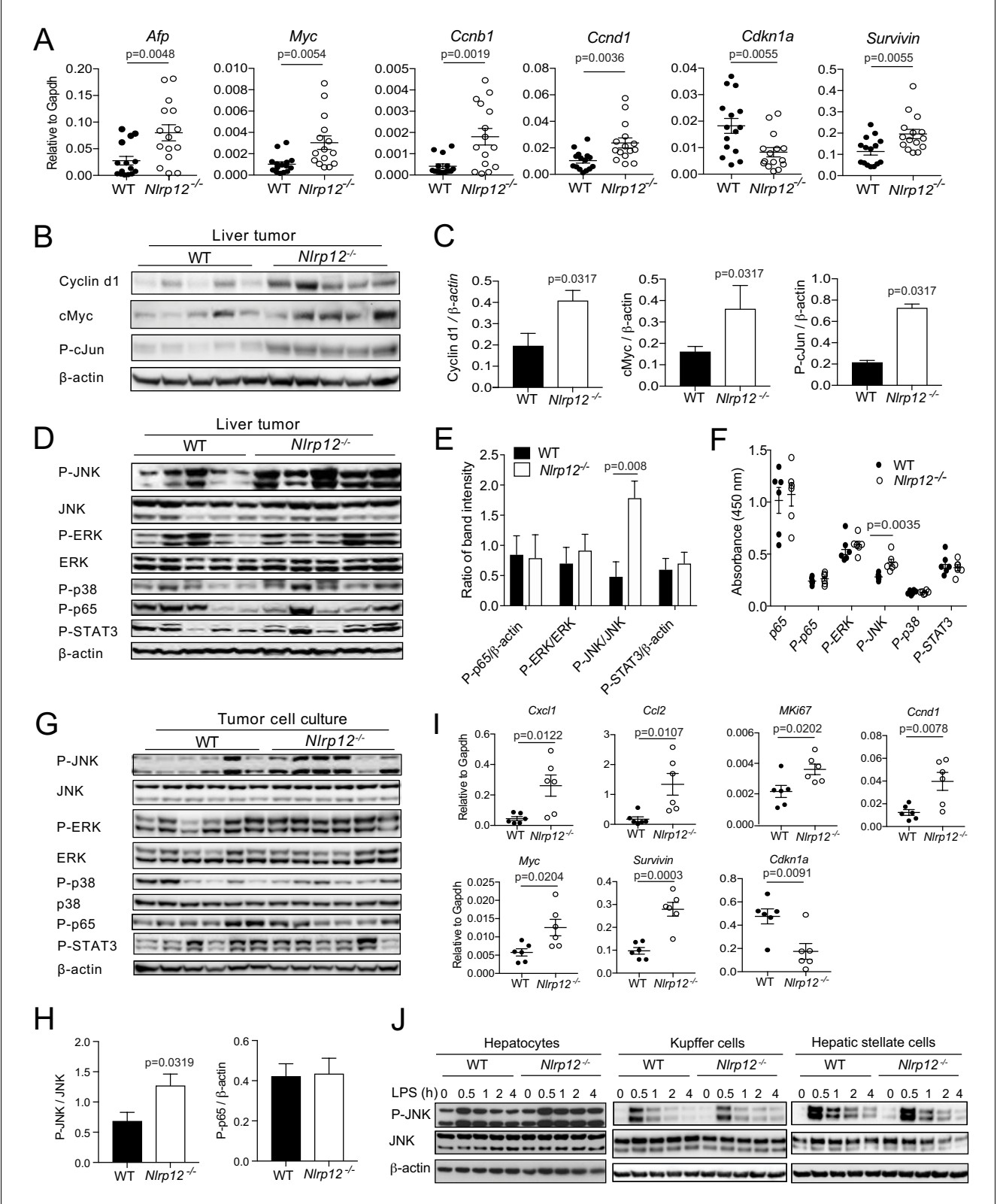

**Figure 4.** NLRP12-deficiency leads to increased expression of proliferative genes and activation of the JNK pathway. WT (n = 15) and *Nlrp12⁻/⁻* (n = 15) mice were injected with DEN (25 mg/kg i.p.) at the age of 14 days and euthanized at 10 months later. (A) Liver tumor tissues were analyzed for the expression of the indicated genes by real-time qPCR. Data represent means ± SEM (n = 15; each sample represents individual mouse). Statistical difference was determined by two-tailed unpaired t-test. (B) Liver tumor lysates were immunoblotted for Cyclin d1, cMyc, and P-cJun. β-actin was used

*Figure 4 continued on next page*

Figure 4 continued

as a loading control. (C) Band intensities of Cyclind1, cMyc, and P-cJun were measured. Data represent means ± SEM (n = 5, each sample represents individual mouse). Statistical difference was determined by two-tailed unpaired t-test. (D) Liver tumor lysates were analyzed for the activation of JNK, ERK, p38, p65, and STAT3 by Western blotting. β-actin was used as a loading control. Each lane represents individual mouse. (E) Band intensities of P-JNK, P-ERK, P-p65, and P-STAT3 shown in D were measured. Data represent means ± SEM (n = 5). Statistical difference was determined by two-tailed unpaired t-test. (F) The levels in p65, P-p65, P-ERK, P-JNK, P-p38, and P-STAT3 in tumor lysates (0.5 mg/ml) from different mice were measured by ELISA. Data represent means ± SEM (n = 6). Statistical difference was determined by two-tailed unpaired t-test. (G) Hepatocytes were isolated from liver tumors and analyzed for the activation of JNK, ERK, p38, p65, and STAT3 by Western blotting. Each lane represents individual mouse sample. (H) Densitometric analysis of P-JNK, and P-p65 immunoreactive bands are shown. Data represent means ± SEM (n = 6). Statistical difference was determined by two-tailed unpaired t-test. (I) RNA isolated from the tumor hepatocytes was analyzed for the expression of chemokines and proliferative genes. Data represent means ± SEM (n = 6, each sample represents individual mouse). Statistical difference was determined by two-tailed unpaired t-test. (J) Hepatocytes, Kupffer cells, and hepatic stellate cells were isolated from liver tumors and stimulated with LPS (1 ug/ml). Activation of JNK was measured by Western blotting.

DOI: https://doi.org/10.7554/eLife.40396.014

The following source data and figure supplements are available for figure 4:

**Source data 1.** NLRP12 suppresses activation of JNK and expression of tumor-promoting molecules during HCC.

DOI: https://doi.org/10.7554/eLife.40396.017

**Figure supplement 1.** NLRP12 downregulates induction of inflammatory molecules in tumor hepatocytes.

DOI: https://doi.org/10.7554/eLife.40396.015

**Figure supplement 1—source data 1.** NLRP12 regulates inflammatory responses in tumor hepatocytes.

DOI: https://doi.org/10.7554/eLife.40396.016

supplement 1C). To understand whether increased HCC susceptibility of Nlrp12$^{-/-}$ mice is due to the defect in the downregulation of PAMPs-mediated HCC pathogenesis, we fed Nlrp12$^{-/-}$ mice with the regular drinking water supplemented with a cocktail of antibiotics that eliminates commensal bacteria (*Rakoff-Nahoum et al., 2004*) and thereby reduces systemic levels of PAMPs (*Seki et al., 2007*). 4 weeks following antibiotics treatment, mice were treated with DEN and HCC development was monitored at 38 weeks of age (*Figure 5A*). Depletion of microbiota was confirmed by colony forming assay and real time PCR analysis of universal bacterial 16S rRNA (data not shown). Antibiotic-treated Nlrp12$^{-/-}$ mice exhibited a profound reduction of tumor burden and expression of AFP after DEN-induced tumorigenesis (*Figure 5B and C*). The expression of proinflammatory cytokines and chemokines was also significantly reduced in livers from antibiotic-treated mice compared to untreated controls (*Figure 5D*). Moreover, the expression of proproliferative molecules including *Myc*, *Ccnd1*, *Ccnb1*, *Survivin*, and *MKi67* in Nlrp12$^{-/-}$ livers was significantly reduced upon antibiotics treatment (*Figure 5E*). These results suggest that gut-derived PAMPs, particularly LPS, promotes HCC development which is downregulated by NLRP12.

LPS is the major cell wall component of gram-negative bacteria and most potent ligand for NF-κB and MAPK activation. To understand whether gram-negative bacteria are abundant in Nlrp12$^{-/-}$ mice, we analyzed gut microbiota composition by 16S rRNA gene sequencing (*Figure 5—figure supplement 1A and B*). The phylum Bacteroidetes, which are gram-negative and most abundant in mouse gut, are equally abundant in WT and Nlrp12$^{-/-}$ mice (*Figure 5—figure supplement 1A*). There was also reduced levels of gram-negative phylum Proteobacteria but a higher abundance of gram-positive phylum Firmicutes in Nlrp12$^{-/-}$ mice (*Figure 5—figure supplement 1A*), suggesting that increased TLR4 responses in Nlrp12$^{-/-}$ HCC may not be due to increased gram-negative bacteria in their gut. Since relative abundance of several bacterial species was different in WT and Nlrp12$^{-/-}$ mice (*Figure 5—figure supplement 1A and B*), we next verified the effect of altered microbiota composition in immune responses in the liver at homeostasis. Hence, we measured the expression of inflammatory molecules, counted the number of immune cells, and analyzed the activation of signaling pathways in healthy WT and Nlrp12$^{-/-}$ livers. Our data show similar levels of *Il6*, *Tnfa*, *Cxcl1*, *Cxcl2,* and *Ccl2* in the liver of healthy WT and Nlrp12$^{-/-}$ mice (*Figure 5—figure supplement 1C*). The number of immune cells including T cells, Kupffer cells, dendritic cells, and neutrophils infiltrated in the liver of healthy WT and Nlrp12$^{-/-}$ mice was comparable (*Figure 5—figure supplement 1D*). There was also no difference in the activation of JNK, NF-κB, ERK, and STAT3 in healthy WT and Nlrp12$^{-/-}$ livers (*Figure 5—figure supplement 1E*). These data suggest that NLRP12 deficiency doesn't have immunomodulatory effect on the liver in the absence of injury or carcinogenesis.

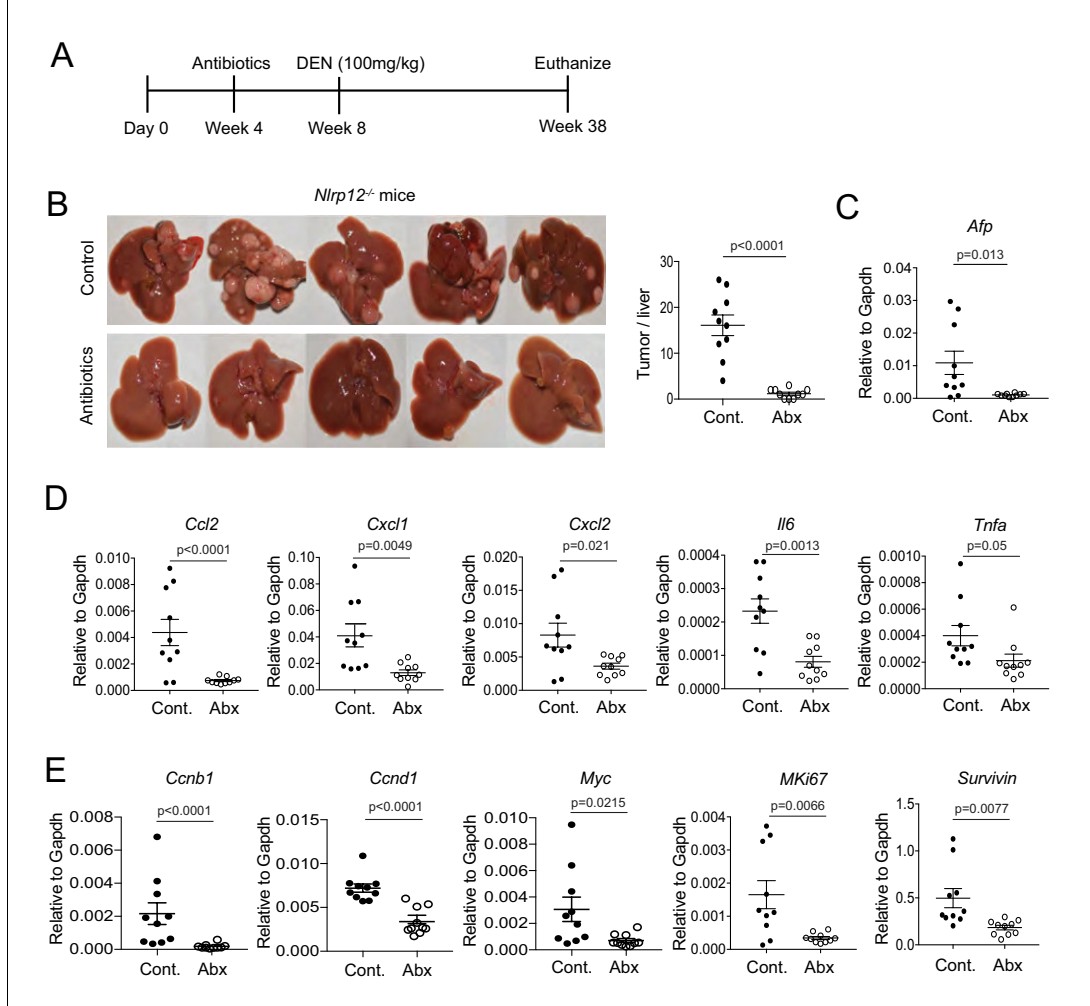

**Figure 5.** NLRP12 suppresses gut microbiota-dependent inflammatory responses and HCC pathogenesis. (**A**) *Nlrp12⁻/⁻* mice were treated with antibiotics in their drinking water starting at 4 weeks after birth and continued until the end of the experiment. Control groups were left untreated. At 4 weeks following antibiotics treatment, all mice (n = 10/group) were injected with DEN (100 mg/kg body weight). (**B**) At 38 weeks, mice were sacrificed and liver tumor development was monitored. Representative images of DEN-treated mouse livers are shown here. Number of tumors were counted. Data represent means ± SEM (n = 10). Statistical difference was determined by two-tailed unpaired t-test. (**C**) The expression of HCC marker AFP in the liver was measured by real-time qPCR. Data represent means ± SEM (n = 10). Statistical difference was determined by two-tailed unpaired t-test. (**D–E**) Liver tissues were analyzed for the expression of cytokines and chemokines (**D**) and pro-proliferative genes (**E**) by real-time qPCR. (**D–F**) Data represent means ± SEM (n = 10). Statistical difference was determined by two-tailed unpaired t-test.

DOI: https://doi.org/10.7554/eLife.40396.018

The following source data and figure supplements are available for figure 5:

**Source data 1.** Measurement of liver tumorigenesis and inflammatory responses following antibiotic treatment.
DOI: https://doi.org/10.7554/eLife.40396.021

**Figure supplement 1.** Altered gut microbiota of *Nlrp12⁻/⁻* mice doesn't influence immune responses in healthy livers.
DOI: https://doi.org/10.7554/eLife.40396.019

**Figure supplement 1—source data 1.** Analyses of gut microbiota composition and inflammatory responses in healthy WT and *Nlrp12-/-* mouse livers.
DOI: https://doi.org/10.7554/eLife.40396.020

## NLRP12 negatively regulates JNK signaling in hepatocytes

Since JNK plays critical roles in hepatocyte physiology and HCC (*Hui et al., 2008*; *Schwabe, 2006*; *Schwabe et al., 2003*), we further investigated whether NLRP12 plays a role as an intrinsic regulator of JNK in hepatocytes. To elucidate the role of NLRP12 in hepatocyte-specific JNK activation, we cultured primary hepatocytes from healthy WT and *Nlrp12⁻/⁻* mouse livers (*Figure 6A* and *Figure 6—*

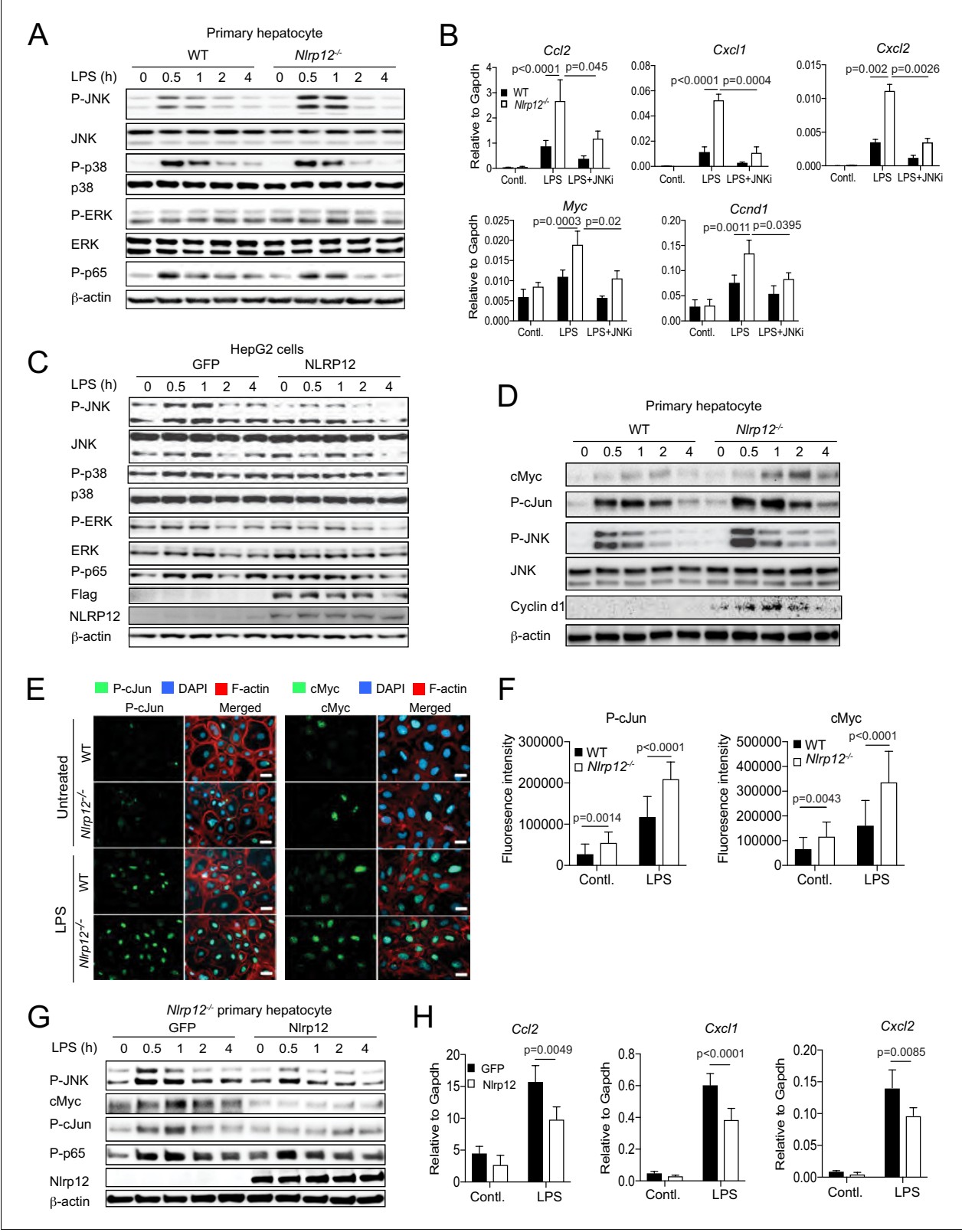

**Figure 6.** NLRP12 downregulates JNK activation in hepatocytes. (**A**) Primary hepatocytes from healthy WT and *Nlrp12*[-/-] mouse livers were isolated and cultured. Hepatocytes were stimulated with LPS for the indicated time and analyzed for the activation of JNK, p38, ERK, and p65 by Western blotting. (**B**) Primary hepatocytes from healthy WT and *Nlrp12*[-/-] mouse livers were stimulated with LPS in the presence of absence of JNK inhibitor. The expression of inflammatory and proliferative molecules was measured by real-time qPCR. Data represent means ± SD (n = 3 replicates). Statistical

*Figure 6 continued on next page*

*Figure 6 continued*

difference was determined by two-tailed unpaired t-test. (C) HepG2 cells stably expressing either GFP or NLRP12 were stimulated with LPS and analyzed for the activation of JNK, p38, ERK, and p65 by Western blotting. (D) Primary hepatocytes isolated from untreated WT and *Nlrp12-/-* mouse livers were stimulated with LPS for the indicated time periods and analyzed for cMyc, P-cJun, Cyclin d1, and P-JNK by Western blotting. (E–F) Primary hepatocytes grown on coverslip were treated with or without LPS for 1 hr and immunostained for P-cJun (green) and cMyc (green). Cellular morphology was visible with filamentous actin (F-actin) staining (red). DAPI (blue) was used for nuclear staining. (F) P-cJun and cMyc fluorescence intensities were measured by Image J software. Data represent means ± SD (n = 20) and is representative of three independent experiments. Statistical difference was determined by two-tailed unpaired t-test. (G–H) *Nlrp12-/-* primary hepatocytes were transiently transfected with either GFP or Nlrp12 constructs followed by stimulation with LPS. The levels of P-JNK, cMyc, P-cJun, and P-p65 were measured by Western blotting (G) and the expression KC (*Cxcl1*), MIP2 (*Cxcl2*), and MCP1 (*Ccl2*) was analyzed by real-time qPCR (H). Data represent means ± SD (n = 3 replicates) and is representative of three independent experiment. Statistical difference was determined by two-tailed unpaired t-test.

DOI: https://doi.org/10.7554/eLife.40396.022

The following source data and figure supplements are available for figure 6:

**Source data 1.** NLRP12 negatively regulates JNK activation and production of inflammatory molecules in hepatocytes.

DOI: https://doi.org/10.7554/eLife.40396.025

**Figure supplement 1.** NLRP12 downregulates multiple JNK activating pathways in hepatocytes.

DOI: https://doi.org/10.7554/eLife.40396.023

**Figure supplement 1—source data 1.** NLRP12 downregulates inflammatory responses in hepatocytes in responses to multiple stimuli.

DOI: https://doi.org/10.7554/eLife.40396.024

*figure supplement 1A*) and stimulated them with LPS. The activation of JNK was enhanced in *Nlrp12-/-* hepatocytes relative to WT; but ERK, p65, and p38 activation was unchanged (*Figure 6A*). Consistent with JNK activation, the expression of *Ccl2, Cxcl1, Cxcl2, Ccnd1, and Myc* was significantly higher in LPS-stimulated *Nlrp12-/-* hepatocytes compared to WT hepatocytes (*Figure 6B*). A pharmacological JNK inhibitor (SP600125) significantly diminished the expression of these molecules, indicating JNK dependency (*Figure 6B*). In addition to LPS, other stimuli such as peptidoglycan (PGN) and TNFα, but not IL-6, induced higher expression of *Cxcl1, Cxcl2, and Ccl2* in *Nlrp12-/-* hepatocytes (*Figure 6—figure supplement 1B*). Consistently, overexpression of NLRP12 in HepG2 cells led to markedly reduced p-JNK levels and subsequent expression of *CXCL1, CXCL2, and CCL2* (*Figure 6C* and *Figure 6—figure supplement 1C*).

Activated JNK phosphorylates cJun and cMyc, and thereby inhibiting their proteasomal degradation (*Alarcon-Vargas and Ronai, 2004*; *Noguchi et al., 1999*). In agreement, the levels of cMyc and P-cJun were markedly higher in LPS-stimulated *Nlrp12-/-* hepatocytes as compared to WT (*Figure 6D*). Immunofluorescence staining showed increased numbers of cMyc and P-cJun-positive cells in *Nlrp12-/-* hepatocytes at basal levels and after stimulation with LPS compared with WT hepatocytes (*Figure 6E and F*). Notably, both cMyc and P-cJun were present in the nucleus consistent with their participation in gene transcription (*Figure 6E*). Increased levels of cMyc and P-cJun in LPS-treated *Nlrp12-/-* hepatocytes (*Figure 6D*) was JNK dependent as inhibition of JNK markedly reduced their levels (*Figure 6—figure supplement 1D*). The role of NLRP12 in regulating JNK, cMyc, and cJun was further confirmed by their rescue after the transient overexpression of NLRP12 in *Nlrp12-/-* hepatocytes (*Figure 6G*). Likewise, the expression of *Cxcl1, Cxcl2, and Ccl2* was suppressed in *Nlrp12-/-* hepatocytes following overexpression of NLRP12 (*Figure 6H*). Finally, we confirmed the role of NLRP12 in the downregulation of JNK by knocking down NLRP12 with CRISPER/Cas9 in HepG2 cells. Suppression of NLRP12 resulted in higher activation of JNK in LPS-stimulated HepG2 cells (*Figure 6—figure supplement 1E*). Taken together, these results suggest that NLRP12 suppresses the expression of inflammatory and oncogenic molecules in hepatocytes via negative regulation of JNK signaling.

## NLRP12 regulates hepatocyte proliferation via JNK

We finally sought to investigate whether NLRP12-mediated regulation of JNK activation affects hepatocyte proliferation. To this end, we isolated and cultured primary hepatocytes from WT and *Nlrp12-/-* mouse livers and followed the cellular proliferation using IncuCyte live cell image analyzer. The proliferation rate of *Nlrp12-/-* hepatocytes was seen significantly higher than WT (*Figure 7A and B*; *Figure 7—figure supplement 1A*), and was further increased by LPS but suppressed by JNK inhibitor (*Figure 7A and B*; *Figure 7—figure supplement 1A*). Consistent with higher proliferation,

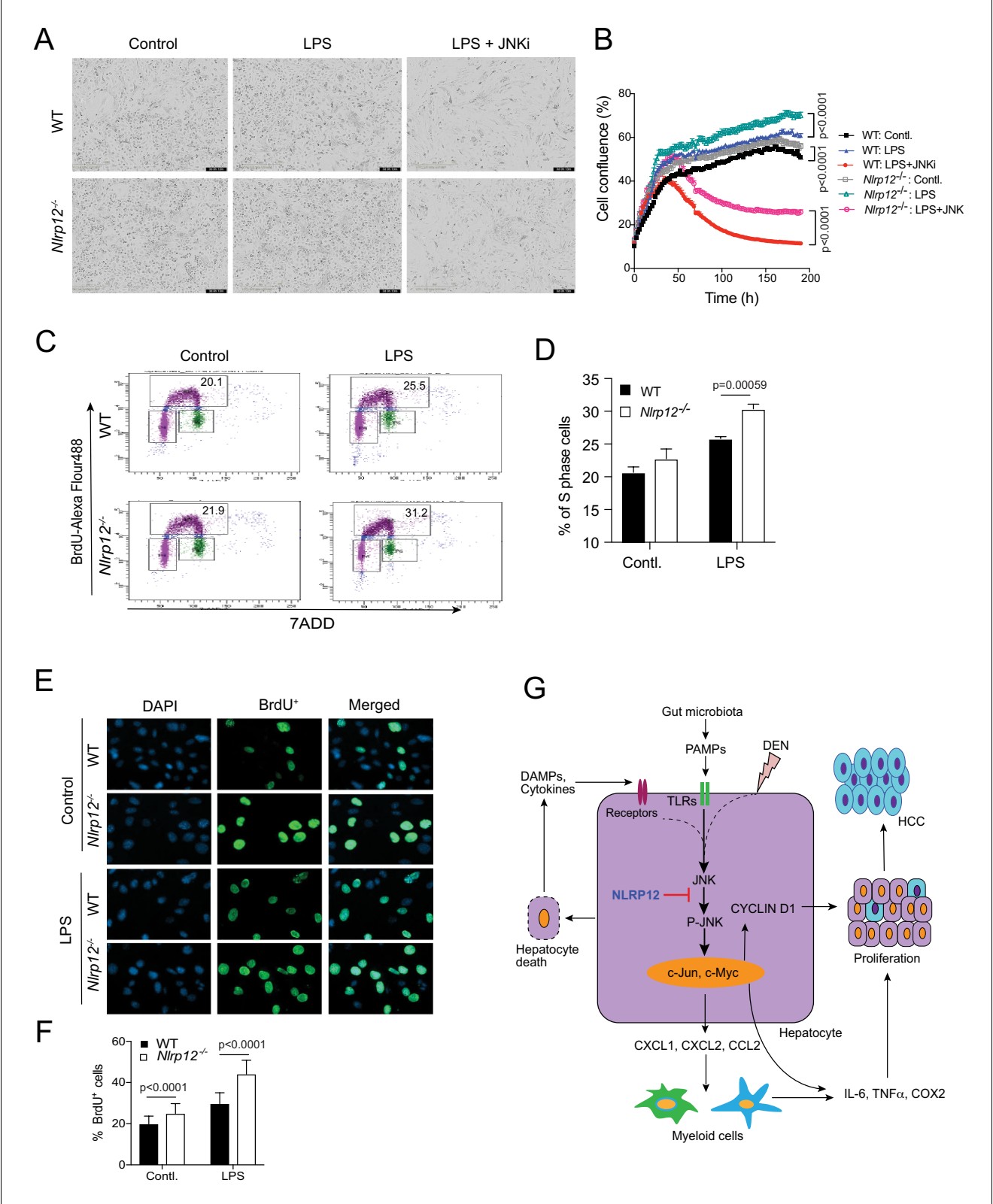

**Figure 7.** NLRP12 regulates hepatocyte proliferation via JNK activation. (**A–B**) Primary hepatocytes from WT and *Nlrp12*<sup>-/-</sup> mouse livers were treated with LPS in the presence or absence of a JNK inhibitor. The proliferation of hepatocytes was monitored in real-time by IncuCyte live cell image analyzer. (**A**) Representative images of hepatocytes captured by IncuCyte are shown. (**B**) The changes in cell confluence were used as a surrogate marker of cell proliferation. Data represent means ± SD (n = 5 replicates) and is representative of three independent experiments. Statistical difference

*Figure 7 continued on next page*

*Figure 7 continued*

was determined by two-tailed unpaired t-test. (**C**) WT and *Nlrp12^-/-* hepatocytes were treated with LPS for 24 hr followed by 1 hr incubation with BrdU. Cells were then immunostained with anti-BrdU antibody and BrdU incorporation (S phase) was analyzed by flow cytometry. (**D**) Percentage of S phase cells as analyzed by flow cytometry was quantitatively analyzed. Data represent means ± SD (n = 3 replicates) and is representative of two independent experiments. Statistical difference was determined by two-tailed unpaired t-test. (**E–F**) Hepatocytes stained with anti-BrdU (green) were observed and counted under the 20X microscopic objective. DAPI (blue) was used for nucleus staining. Data represent means ± SD (n = 30 replicates) and is representative of two independent experiments. Statistical difference was determined by two-tailed unpaired t-test. (**G**) The proposed mechanism of NLRP12-mediated regulation of HCC.

DOI: https://doi.org/10.7554/eLife.40396.026

The following source data and figure supplements are available for figure 7:

**Source data 1.** Assesment of the role of Nlrp12 in hepatocyte proliferation.

DOI: https://doi.org/10.7554/eLife.40396.029

**Figure supplement 1.** JNK-dependent hepatocyte proliferation is regulated by NLRP12.

DOI: https://doi.org/10.7554/eLife.40396.027

**Figure supplement 1—source data 1.** Analysis of Ki67-positive cells in LPS-stimulated hepatocytes.

DOI: https://doi.org/10.7554/eLife.40396.028

*Nlrp12^-/-* hepatocytes with or without LPS stimulation showed markedly higher Ki67 staining (***Figure 7—figure supplement 1B and C***). To observe cell cycle progression in WT and *Nlrp12^-/-* hepatocytes, we incubated WT and *Nlrp12^-/-* hepatocytes with BrdU, which incorporates into newly synthesized DNA at S phase. BrdU-positive cells were analyzed by flow cytometry and microscopy. There was a trend of increased BrdU-positive S phase hepatocytes in the *Nlrp12^-/-* group compared to WT which was further increased significantly with LPS stimulation (***Figure 7E and F***). Similarly, *Nlrp12^-/-* hepatocytes showed markedly higher BrdU immunoreactivity upon stimulation with LPS (***Figure 7E and F***). These results indicate that NLRP12 controls hepatocytes proliferation via regulation of JNK.

In summary, our data suggest that NLRP12 plays a central role in hepatocyte function by dampening the expression of cytokines, chemokines, growth factors, and oncogenes via negative regulation of the JNK pathway (***Figure 7G***). NLRP12 deficiency, therefore, enhances hepatocyte-specific inflammatory and proliferative responses during DEN-induced liver injury, leading to higher incidence of HCC (***Figure 7G***).

## Discussion

Despite the high mortality rate, effective treatment options for HCC remain limited, emphasizing the importance of finding therapeutic measures to impede carcinogenesis. It is commonly viewed that HCC is initiated and promoted with consistent and nonspecific activation of the immune system in the liver (***El-Serag and Rudolph, 2007***; ***Farazi and DePinho, 2006***). Gut microbiota have recently emerged as a critical pathogenic trigger for HCC (***Dapito et al., 2012***; ***Yoshimoto et al., 2013***). TLR4-mutant mice, which lack responsiveness to LPS, are resistant to liver fibrosis, cirrhosis, and HCC (***Dapito et al., 2012***; ***Seki et al., 2007***). Although the liver is not in a direct contact with gut microbiota, microbes and their PAMPs are transported to the liver through the hepatic portal vein after crossing the gut epithelial barrier (***Son et al., 2010***). Cirrhosis facilitates intestinal barrier permeability and cirrhotic patients are vulnerable for bacterial infection which increases mortality (***Campillo et al., 1999***; ***Cirera et al., 2001***; ***Fukui et al., 1991***; ***Pascual et al., 2003***; ***Rutenburg et al., 1957***; ***Yoneyama et al., 2002***). However, the precise mechanism of gut microbiota-mediated augmentation of HCC pathogenesis remains to be determined. Here we show that NLRP12 is a potent molecular checkpoint for gut microbiota-dependent inflammation and carcinogenesis in the liver.

The pathogenesis of HCC is a complex process because of its genetic heterogeneity. Genome sequencing and transcriptomics analyses have identified *CTCNB1, WNT, AXIN, TP53, CCND1, CDKN2A, TERT, ARID1A, and ARID2* as major genes whose alterations cause HCC induction (***Huang et al., 2012***; ***Marquardt et al., 2015***). Since somatic mutations in these driver oncogenes and tumor suppressors occur in the liver with the exposure of carcinogen, chronic inflammation, and oxidative stress, these pathological stimuli also alter many other genes (***Marquardt et al., 2015***).

Genome sequencing of tumors from an HBV-infected HCC individual has identified more than 11,000 somatic substitutions (*Totoki et al., 2011*). Significant genetic alterations in NLRs have been identified in a genome sequence study (*Everson et al., 2013*). Although at low frequency, mutations in passenger genes promote tumorigenesis in multiple ways including hyperactivation of inflammatory signaling pathways such as NF-κB, MAPK, AKT, and JAK/STAT (*Nault and Zucman-Rossi, 2011*). As a regulator of NF-κB and MAPK pathways, therefore, *NLRP12* mutations may play a critical role in HCC pathogenesis. Increased HCC susceptibility of *Nlrp12*[-/-] mice suggests that Nlrp12 contributes to the suppression of HCC, which is also supported by human HCC genomics data showing *NLRP12* mutations in 1–2% HCC. Notably, like many other cancer-related genes, amplification of NLRP12 as seen in less than 0.5% HCC cases may also contribute to HCC. It is possible that higher NLRP12 activity suppresses JNK to a level that promotes HCC. In fact, JNK and NF-κB pathways play both tumor promoting and tumor suppressive roles (*Das et al., 2011*; *Hui et al., 2008*; *Inokuchi et al., 2010*; *Luedde et al., 2007*; *Maeda et al., 2005*; *Sakurai et al., 2006*).

Growing evidence implicates the role of NLRP12 in diverse pathophysiological conditions. Variants of *NLRP12* have been linked to autoinflammatory diseases, including periodic fever syndrome, atopic dermatitis, and arthritis (*Borghini et al., 2011*; *Jéru et al., 2008*; *Jéru et al., 2011*). Experimental studies demonstrated that NLRP12 protects mice from colitis, colorectal tumorigenesis, and experimental autoimmune encephalomyelitis (EAE) (*Allen et al., 2012*; *Lukens et al., 2015*; *Zaki et al., 2014*; *Zaki et al., 2011*). Mechanistically, NLRP12 regulates these diseases via down-regulation of the canonical and non-canonical NF-κB pathways (*Allen et al., 2012*; *Zaki et al., 2011*). However, NLRP12-mediated regulation of NF-κB under different pathophysiological contexts appears to be cell type-dependent. For example, NLRP12-mediated inhibition of NF-κB in myeloid cells contributes to protection against intestinal inflammation and tumorigenesis, in T cells it is implicated in experimental autoimmune encephalomyelitis (*Lukens et al., 2015*), and in osteocytes it is involved in osteoclast differentiation (*Krauss et al., 2015*). This study shows that NLRP12 is expressed in liver parenchymal and non-parenchymal cells and suppresses HCC by preventing JNK signaling, particularly in the hepatocyte. Thus, our findings elucidate a novel function of NLRP12 in the liver and underscore its versatile physiological functions.

Although we could not find a difference in the activation of NF-κB and ERK between WT and *Nlrp12*[-/-] HCC, the involvement of NLRP12 in the downregulation of these pathways in myeloid and T cells during HCC and a contribution of such processes in the suppression of HCC cannot be completely excluded. It might be possible that NLRP12 suppresses NF-κB and ERK in immune cells during acute but not in a chronic inflammatory environment. Thus, NLRP12-mediated downregulation of NF-κB and ERK activation as well as inflammatory responses in immune cells may occur during the early phase of HCC development. The contribution of the immune cell-specific function of NLRP12 in the suppression of HCC needs to be further investigated using Nlrp12 conditional knockout mice in future studies.

In this study, we show that NLRP12 attenuates HCC development though multiple mechanisms. First, NLRP12 protects the liver from injury and hepatocyte death. The liver is a unique organ with regenerative capacity. Hepatocyte death due to cytotoxic insult or inflammatory responses triggers rapid proliferation to compensate for apoptotic or necrotic hepatocyte cell death (*Sakurai et al., 2008*). Such a rapid cell cycle progression in an inflammatory environment results in accumulation of mutations leading to neoplastic transformation of hepatocytes. Secondly, NLRP12 downregulates hepatocyte-specific expression of cytokines such as IL-6 and TNFα which promote proliferation and tumor growth, and chemokines CXCL1, CXCL2 and CCL2, which enhance inflammatory cell infiltration and thereby augment inflammation in the tumor milieu (*Kamata et al., 2005*; *Maeda et al., 2005*; *Sakurai et al., 2008*). Tumor infiltrated immune cells, particularly macrophages, play critical roles in HCC induction and progression by producing many tumor-promoting factors such as iNOS, Cox2, and ROS (*Capece et al., 2013*; *Grivennikov et al., 2010*). Third, NLRP12 inhibits the expression of cJun and cMyc, which are major oncogenes seen in HCC (*Lin et al., 2010*). These oncogenic transcription factors regulate the expression of molecules involved in proliferation and cell cycle progression, such as Ccnb1, Ccnd1, and Cdkn1a. Consistently, we observed higher induction of *Ccnb1* and *Ccnd1* but reduced expression of *Cdkn1a* in *Nlrp12*[-/-] hepatocytes and HCC. Notably, NLRP12-mediated regulation of these processes occurs within the context of PAMP-stimulation as inhibition of gut microbiota abolishes the HCC susceptibility of *Nlrp12*[-/-] mice.

Previous studies implicated the NF-κB pathway in NLRP12-mediated regulation of inflammatory disorders. However, for the first time, we show that NLRP12 regulates JNK activation in hepatocytes. JNK has critical functions in liver physiology and contributes to HCC pathogenesis (*Das et al., 2011*; *Hui et al., 2008*; *Sakurai et al., 2006*; *Schwabe, 2006*). JNK is highly activated in human HCC and mouse deficient in JNK1 develop reduced tumor burden in the liver following DEN treatment (*Hui et al., 2008*; *Sakurai et al., 2006*). Similar to our observation that NLRP12 regulates hepatocyte death and proliferation, previous studies relate JNK1-dependent HCC pathogenesis to hepatocyte death and proliferation (*Hui et al., 2008*; *Sakurai et al., 2006*). The mechanism of JNK-mediated regulation of cellular proliferation involves activation of c-Jun, JunD, and cMyc, depending on the cell type and stimuli (*Alarcon-Vargas and Ronai, 2004*; *Bogoyevitch and Kobe, 2006*). The role of cJun and cMyc in hepatocellular carcinogenesis is well documented (*Eferl et al., 2003*). In agreement, we observed that increased HCC pathogenesis in *Nlrp12*$^{-/-}$ mice is associated with increased cMyc and cJun activity. Moreover, NLRP12-deficiency links JNK activation to higher inflammatory responses during DEN-induced liver tumorigenesis. Through in vitro biochemical studies using primary hepatocytes, we clearly demonstrated that NLRP12 regulates JNK activation in the hepatocyte, particularly in the context of TLR stimulation. It is worthwhile to mention that higher JNK activation in *Nlrp12*$^{-/-}$ liver is only seen during HCC; JNK activation as well as inflammatory responses in the liver of healthy WT and *Nlrp12*$^{-/-}$ mice were comparable. It seems that tumor induction sensitizes hepatocytes to TLR ligands, leading to increased activation of JNK. It is also possible that HCC development increases intestinal epithelial barrier permeability, allowing increased translocation of gut-derived PAMPs into the liver. Higher levels of plasma endotoxin and bacterial translocation are seen in patients with liver cirrhosis (*Campillo et al., 1999*; *Cirera et al., 2001*; *Fukui et al., 1991*). Thus, NLRP12 may play a functional role in suppressing JNK activation in the hepatocyte in the context of liver injury, fibrosis, and tumorigenesis.

In summary, this study demonstrates that the innate pathogen sensor NLRP12 is a molecular checkpoint for HCC. NLRP12 suppresses HCC by attenuating JNK-mediated inflammatory and proliferative responses in the hepatocytes, particularly in the context of stimulation with microbial pattern molecules. Thus, this study suggests that inducing or activating NLRP12 or its downstream signaling could be potential therapeutic options for HCC. Considering the critical role of gut microbiota and TLR pathways in inflammatory liver disorders (*Guo and Friedman, 2010*; *Seki et al., 2007*), future studies should investigate the role of NLRP12 in liver fibrosis, cirrhosis, and non-alcoholic fatty liver syndrome using appropriate animal models.

## Materials and methods

### Key resources table

| Reagent type or resource | Designation | Source or reference | Identifiers | Additional information |
|---|---|---|---|---|
| Genetic reagent (M. musculus) | C57BL/6J | Jackson Lab | RRID:MGI:3028467 JAX:000664 | |
| Genetic reagent (M. musculus) | *Nlrp12*$^{-/-}$ C57BL/6J | PMID:12563287 | RRID: MGI:2676630 | |
| Cell line (*Homo Sapiens*) | HepG2 | ATCC | RRID:CVCL_0027 | |
| Recombinant DNA reagent | pcDNA4/TO | Invitrogen | Cat#V1020-20 | |
| Recombinant DNA reagent | pcDNA4/TO-Nlrp12 | This paper | | Full length Nlrp12 (mouse) cDNA was cloned into pcDNA4/TO |
| Recombinant DNA reagent | pcDNA4/TO-NLRP12 | This paper | | Full length NLRP12 (human) cDNA was cloned into pcDNA4/TO |
| Recombinant DNA reagent | NLRP12 sgRNA CRISPR/Cas9 All-in-One Lentivector (Human) | ABM Inc | Cat # K1434706 | |

*Continued on next page*

*Continued*

| Reagent type or resource | Designation | Source or reference | Identifiers | Additional information |
|---|---|---|---|---|
| Recombinant DNA reagent | Scrambled sgRNA CRISPER/Cas9 All-in-one Lentivector | ABM Inc | Cat # K010 | |
| Antibody | Rabbit polyclonal anti-NLRP12 | Aviva Systems Biology | RRID:SCR_001456 Cat # OAAB04256 | WB: 1:500 |
| Antibody | Rabbit monoclonal anti-p44/42 (Erk1/2) | Cell Signaling | RRID:AB_390779 Cat # 4695, | WB: 1:2000 |
| Antibody | Rabbit monoclonal anti-Phospho-SAPK/JNK | Cell Signaling | RRID:AB_823588 Cat # 4668 | WB: 1:1000 |
| Antibody | Rabbit monoclonal anti-p38 MAPK | Cell Signaling | RRID:AB_10999090 Cat # 8690 | WB: 1:1000 |
| Antibody | Rabbit monoclonal anti-Phospho-AKT | Cell Signaling | RRID:AB_2315049 Cat # 4060 | WB: 1:1000 |
| Antibody | Rabbit monoclonal anti-cMyc | Cell Signaling | RRID:AB_1903938 Cat # 5605 | WB: 1:1000, IF:1:100 |
| Antibody | Mouse monoclonal anti b-actin | Sigma | RRID:AB_476697. Cat # A2228 | WB: 1:10000 |
| Antibody | Rabbit polyclonal anti-SAPK/JNK | Cell Signaling | RRID:AB_2250373 Cat # 9252 | WB: 1:5000 |
| Antibody | Rabbit monoclonal anti-Phospho-p44/42 (ERK1/2) | Cell Signaling | RRID:AB_2315112 Cat # 4370 | WB: 1:2000 |
| Antibody | Rabbit monoclonal anti-Phospho-cJun | Cell Signaling | RRID:AB_2129575 Cat # 3270 | WB: 1:1000, IF:1:100 |
| Antibody | Rabbit monoclonal anti-Phospho-p38 MAPK | Cell Signaling | RRID:AB_331762 Cat # 9215 | WB: 1:1000 |
| Antibody | Rabbit monoclonal anti-Phospho-STAT3 | Cell Signaling | RRID:AB_2491009 Cat # 9145 | WB: 1:1000 |
| Antibody | Rabbit monoclonal Phospho-NF-kB p65 | Cell Signaling | RRID:AB_331284 Cat # 3033 | WB: 1:2000 |
| Antibody | Rabbit monoclonal anti-Cyclin d1 | Cell Signaling | RRID:AB_2259616 Cat # 2978 | WB: 1:1000 |
| Antibody | Rabbit polyclonal anti-Akt | Cell Signaling | RRID:AB_329827. Cat # 9272 | WB: 1:1000 |
| Antibody | Rabbit monoclonal anti-Ki67 | abcam | RRID:AB_302459 Cat # ab16667 | IF:1:100 |
| Antibody | Rat monoclonal anti-F4/80 (Clone CI:A3-1) | BioRad | RRID:AB_323806 Cat # MCA497GA | IF:1:100 |
| Antibody | Mouse monoclonal anti-FlagM2 | Sigma | RRID:AB_262044 Cat # F1804 | WB: 1:10000 |
| Antibody | Mouse monoclonal anti-α-BrdU | Cell signaling | RRID:AB_10548898 Cat # 5292 | IF: 1:200 |
| Antibody | Rat monoclonal anti-CD16/CD32 (clone 2.4G2) | BioLegend | RRID:AB_394656 clone 2.4G2 | 1 µg/ $10^6$ cells |
| Antibody | Mouse monoclonal Pacific Blue anti-CD45.2 Antibody | BioLegend | RRID:AB_492873 Cat # NC0123437 | 1:100 |
| Antibody | Rat monoclonal PerCP-Cyanine5.5 Anti-Human/Mouse CD11b (M1/70) | Tonbo Bioscience | RRID:AB_2621885 Cat# 65–0112 | 1:100 |

*Continued on next page*

*Continued*

| Reagent type or resource | Designation | Source or reference | Identifiers | Additional information |
|---|---|---|---|---|
| Antibody | Rat monoclonal APC Anti-Mouse F4/80 Antigen (BM8.1) | Tonbo Bioscience | RRID:AB_2621602 Cat # 20–4801 | 1:100 |
| Antibody | Rat monoclonal In Vivo Ready Anti-Mouse Ly-6G (Gr-1) (RB6-8C5) | Tonbo Bioscience | RRID:AB_2621463 Cat # 40–5931 | 1:100 |
| Antibody | Monoclonal Anti-CD11c conjugated with PE | Tonbo Bioscience | RRID:AB_2621747 Cat # 50–0114 | 1:100 |
| Antibody | Monoclonal PE/Cy7 anti-mouse TCR β chain | Biolegend | RRID:AB_893627 Cat # 109221 | 1:100 |
| Chemical compound | Zeocin | Invivogen | Cat # ant-zn-1 | (100 µg/ml) |
| Chemical compound | Lipofectamine 3000 | Thermo Fisher | Cat # L3000015 | |
| Chemical compound | Mycoplasma Kit | Sigma | Cat#11663925910 | |
| Chemical compound | Ultrapure *Escherichia coli*-derived LPS | Invivogen | Cat # tlrl-smlps | |
| Chemical compound | PGN | Invivogen | Cat # tlrl-pgnsa | |
| Recombinant protein | Recombinant Human IL-6 | Peprotech | Cat # 200–06 | |
| Recombinant protein | Recombinant Murine TNFα | Peprotech | Cat # 315-01A | |
| Commercial kit | In Situ Cell Death Detection Kit-Fluorescein | Roche | Cat # 11684795910 | |
| Commercial kit | Pathscan Inflammation Multi-target ELISA kit | Cell Signaling Technology | Cat # 7276 | |
| Commercial kit | Pathsan Phospho-p44/42 MAPK ELISA kit | Cell Signaling Technology | Cat # 7177C | |
| Software, algorithm | BD FACS Diva software | BD Bioscience | | |
| Software, algorithm | Flowjo v10 | Treestar, Inc | RRID:SCR_008520 | |
| Software, algorithm | GraphPad Prism | graphpad.com | RRID:SCR_002798 | |
| Software, algorithm | QIIME 1.8.0 | Qiime.org | RRID:SCR_008249 PMC3156573 | |

## Mice

Wild-type (C57BL6/J) mice were purchased from Jackson Laboratory. *Nlrp12-/-* was generated by Millenium Pharmaceuticals and backcrossed for 10 generations with C57BL6/J mice. All mice were bred and maintained in a specific pathogen free (SPF) facility at UT Southwestern Medical Center. Unless otherwise stated, mice of different genetic backgrounds were housed in separate cages, maintained in same animal room, and used for in vivo and in vitro experiments. This study was performed under the protocol #2016–101683 which was approved by the Institutional Animal Care and Use Committee (IACUC). All animal experiments were conducted in accordance with the IACUC guidelines and the National Institutes of Health Guide for the Care and Use of Laboratory Animals. All experiments were conducted with age and sex-matched mice and treatment groups were allocated randomly.

## Cell culture

HepG2 cell line was collected from ATCC. The cells were authenticated by UT Southwestern genomics core facility using Short Tandem Repeat (STR) DNA profiling. The cells were cultured in DMEM with 10% FBS plus 1X Pen/Strep (Thermo Scientific). Cells were tested for mycoplasma contamination using mycoplasma testing kit (Sigma).

## Induction of hepatocellular carcinoma (HCC)

Mice were injected with DEN (25 mg/kg i.p.) intraperitoneally (i.p.) at day 14 postpartum. 10 months following DEN injection, mice were euthanized with $CO_2$. In another approach, HCC was induced by the combination of DEN (25 mg/kg i.p.) given at day 14 postpartum followed by weekly injections of $CCl_4$ (0.5 ml/kg i.p., dissolved in corn oil) for 8 weeks starting at 10 weeks after birth. Mice were sacrificed at the age of 6 months.

In antibiotics study, 4-week-old mice were treated with a combination of antibiotics including ampicillin (1 g/L), neomycin (1 g/L), metronidazole (1 g/L), streptomycin (1 mg/L) and vancomycin (500 mg/L) in drinking water. At 8 weeks of age, mice were injected with DEN (100 mg/kg, i.p.). Antibiotics treatment was continued until the end of the study. Elimination of gut microbiota was confirmed by measuring eubacterial 16S rDNA by real-time PCR and culturing of fecal homogenates on brain heart infusion (BHI) agar.

## Histopathology and immunohistochemistry

Liver tissue samples were fixed in 4% paraformaldehyde and embedded in paraffin. The tissue sections were stained with hematoxylin and eosin (H and E). Histopathological scoring was done in a blinded fashion by a pathologist. Steatosis was scored as: 0: 0% to 5%; 1: 5% to 33%; 2: 33% to 66%; and 3: greater than 66% of liver shows steatosis. Inflammation was scored as: 0: no foci/20x field; 1:<2 foci/20x field; 2: 2–4 foci/20x field; and 3:>4 foci/20x field. Fibrosis was defined as: 0: no fibrosis; 1: mild fibrosis, focally or extensively present; 2: moderate fibrosis, extensive periportal fibrosis; 3: severe fibrosis, extensive bridging fibrosis; and 4: cirrhosis. HCC was scored as a percentage of area covered by tumor.

For immunohistochemistry, 4% paraformaldehyde-fixed and paraffin-embedded tissue sections were de-paraffinized and hydrated through decreasing concentrations of ethanol. Antigen retrieval was done in 10 mM sodium citrate solution (pH 6.0) for 20 min at 95°C. Tissue sections were blocked with 5% goat serum for 30 min and stained for Ki67 using rabbit anti-Ki67 (ab16667; Abcam), anti-F4/80 (Clone CI:A3-1; MCA497GA, BioRad), and anti-cleaved caspase-3 (Asp175; 5A1; 9664; Cell Signaling Technologies). After overnight incubation at 4°C, the tissue sections were washed three times and incubated with HRP-conjugated anti-rabbit antibody for 1 hr at room temperature. The images were taken using bright field microscopy.

## Immunofluorescent staining

For immunofluorescent staining of hepatocytes, cells were seeded on cover slips and fixed in 4% paraformaldehyde for 15 min at RT, washed with PBS 3 times for 5 min each and blocked with PBS containing 5% goat serum and 0.3% Triton-X100 for 1 hr. Cells were then incubated with primary antibodies against Ki-67 (ab16667; Abcam), albumin (NB600-41532, Novus), phospho-cJun (Ser73, D47G9, 3270, Cell Signaling), cMyc (D84C12, 5605, Cell Signaling) overnight at 4°C. After washing in PBS, cells were incubated with anti-rabbit secondary antibodies conjugated with Alexa Fluor 564 (Invitrogen), Alexa Fluor 488 (Invitrogen), or FITC (Sigma), for 1 hr at RT. The cells were washed in PBS and incubated with Flash Phalloidin Red 594 antibody (424203, Biolegend) for 30 min at RT. Following three washes in PBS, cells were mounted with mounting media containing DAPI. Images were taken by fluorescence microscope (Zeiss).

## Hepatocytes isolation and culture

Hepatocytes were isolated from mouse livers by a collagenase perfusion technique. In brief, after the left ventricle was cannulated and right atrium was cut, the liver was perfused with 20 ml PBS followed by 30 ml HBSS (without $Mg^{2+}$ or $Ca^{2+}$) supplemented with 0.2 mM EDTA at 10 ml min$^{-1}$ speed through the left ventricle. Next, the liver was perfused with 20 ml collagenase type IV (Sigma Aldrich) containing medium (0.5 mg/ml collagenase type IV in DMEM supplemented with 5 mM

HEPES, Penicillin/Streptomycin). The liver was dissociated and the liver suspension was passed through a 70 µm sterile filter. The hepatocytes were separated from non-parenchymal cells by low-speed centrifugation (50x*g* for 5 min). The hepatocytes were further purified using Percoll gradient separation. The living hepatocytes were counted using Trypan blue and cultured on collagen-coated plate having DMEM medium supplemented with 10% FBS, 1x Penicillin/Streptomycin, 1x Insulin (sigma), and EGF (40 ng/ml). For priming, hepatocytes were cultured in collagen-coated 6- or 12-well cell culture plates overnight and stimulated with TLR ligands: ultrapure *Escherichia coli*-derived LPS (Invivogen), PGN (Invivogen), IL-6 (Peprotech), or TNFα (Peprotech).

## Isolation and culture of hepatic stellate cells and Kupffer cells

The liver was dissociated as described above and the liver suspension was passed through a 70 µm sterile filter. The hepatocytes were separated from non-parenchymal cells by low-speed centrifugation (50x*g* for 5 min) as described above. The supernatant was collected and centrifuged at 640xg for 10 min and resuspended in washing buffer followed by pass through a 70 µm sterile filter. The pellet was resuspended in 10 ml of 35% Percoll (GE Healthcare, Pittsburgh, PA, USA) with an overlay of 1 ml PBS. After centrifugation at 1130xg for 30 min, hepatic stellate cells are in the layer located between the PBS and 35% Percoll. Kupffer cells were separated from the remainder of non-parenchymal cells by Percoll gradient centrifugation at 800xg for 30 min. After centrifugation, Kupffer cells are in the layer located between the 70% and 40% Percoll. The hepatic stellate cells were cultured in RPMI 1640 medium supplemented with 10% FBS, 1x Penicillin/Streptomycin, 1x Insulin (sigma). The Kupffer cells were cultured on six-well plate having DMEM medium supplemented with 10% FBS, 1x Penicillin/Streptomycin. For priming, the cells were cultured in 6- or 12-well cell culture plates overnight and stimulated with ultrapure *Escherichia coli*-derived LPS (Invivogen).

## Flow cytometric analysis of liver non-parenchymal cells

The harvested livers were cut into small pieces (1–2 mm) and digested with liver digestion media (0.5 mg/ml Collagenase Type IV, 5 mM HEPES, and 1X Penicillin-streptomycin in HBSS) for 30 min at 37°C while shaking in a water bath. HBSS media supplemented with 2% heat-inactivated FBS and 5 mM EDTA was added into the digested liver to stop the digestion followed by filtering through a 70 µm sterile filter. The filtered liver suspension was centrifuged at 450xg for 8 min at 4°C. The cell pellet was resuspended in ice-cold RBC lysis buffer, kept on ice for 3 min and the lysis buffer was neutralized by adding RPMI media supplemented with 5% FBS. Following centrifugation at 450xg for 8 min at 4°C, the cell pellet was washed three times with RPMI media supplemented with 5% FBS. The cell pellet was resuspended in RPMI media supplemented with 5% FBS and the living cells were counted using Trypan blue. $1 \times 10^6$ live cells per sample were stained with Live/Dead fixable yellow dead cell stain kit (Life Technologies) according to the manufacturer's instructions. Cells were then blocked with Fc Block (anti-mouse CD16/CD32; clone 2.4G2; 1ug/$10^6$ cells; BioLegend) in FACS Buffer (2% FBS in PBS) for 20 min at 4°C. Following blocking, cells were stained for 30 min at 4°C with anti-CD45.2 conjugated with Pacific blue (1:100 dilution, NC0123437, BioLegend), anti-CD11b conjugated with PerCP-Cy5.5 (Clone M1/70, 1:100 dilution, 65–0112 U025, Tonbo Bioscience), anti-F4/80 conjugated with APC (Clone BM8.1, 1:100 dilution, 20–4801 U025, Tonbo Bioscience), anti-Ly6G (Gr.1) conjugated with FITC (1:100 dilution, 40–5931 U100, Tonbo Bioscience), anti-CD11c conjugated with PE (1:100 dilution, 50–0114 U025, Tonbo Bioscience), and anti-TCRb conjugated with PE-Cy7 (1:100 dilution, 109221, Biolegend). After staining, cells were washed and fixed with 4% paraformaldehyde in PBS. Data were acquired with a BD LSRII Fortessa flow cytometer using BD FACSDiva software (BD Bioscience). Compensation was performed on the BD LSRII flow cytometer at the beginning of each experiment. Data were analyzed by using Flowjo v10 (Treestar, Inc).

## In vivo BrdU incorporation assay

BrdU incorporation assay was done using BrdU In-Situ Detection Kit (550803, BD Pharmingen) according to the manufacturer's instructions. In brief, 2 hr prior to euthanasia mice were injected intraperitoneally with BrdU (B5002, Sigma Aldrich), at a dose of 50 mg/kg body weight of mice. Tissue samples were fixed in 4% paraformaldehyde and embedded in paraffin. The paraffin-embedded liver tissue sections were de-paraffinized by washing with xylene 2 times for 5 min each time at room temperature. The tissue sections were dehydrated by incubation in 100% ethanol 2 times for 5 min

followed by once in 95% ethanol for 3 min at room temperature. The tissue sections were treated with 0.3% $H_2O_2$ to block endogenous peroxidase, followed by antigen retrieval with 'BD Retrievagen A' (#550803, BD Pharmingen) in a microwave oven to 89°C for 10 min. The tissue sections were incubated in biotinylated anti-BrdU antibody (#550803, BD Pharmingen) at 1:10 in diluent buffer for 1 hr at room temperature, and then in HRP-conjugated streptavidin for 1 hr at room temperature. The tissue sections were stained with DAB substrate solution followed by hematoxylin counterstaining. Slides were examined by bright field microscopy.

## In vitro proliferation assay using BrdU incorporation

For measuring proliferation using BrdU incorporation in vitro, cells were incubated with BrdU (10 µM) in culture medium for 2 hr at 37°C with 5% $CO_2$. The cells were then fixed in 70% ethanol for 5 min, treated with 1.5 N HCl for 30 min at RT followed by washing with PBS. Following blocking with 5% normal goat serum in PBS plus 0.3% Triton-X100 for 1 hr the cells were incubated with anti α-BrdU antibody (Bu20a, 5292, Cell signaling) overnight at 4°C. After washing in PBS, the cells were treated with anti-mouse antibody conjugated with Alexa flour 488 for 2 hr at RT. Finally, the cells were washed with PBS and mounted with mounting media having DAPI. Images were taken by fluorescence microscope (Zeiss). For flow cytometric analysis, the cells were fixed in ice-cold 70% ethanol, followed by treatment with 1.5 HCl plus 0.5% Triton X-100 to permeabilize the cell and denature the DNA. The cells were neutralized with 0.1 M sodium tetraborate followed by rinse with PBS and incubated with anti α-BrdU antibody (Bu20a, 5292, Cell signaling). Anti-mouse antibody conjugated with Alexa flour 488 was used for counterstaining. DNA was stained with 2.5 µg/ml 7-AAD in the presence of 10 µg/mL RNase A. Samples were analyzed on a BD LSRII Fortessa flow cytometer using BD FACSDiva software (BD Biosciences).

## Real-time cell proliferation assay

The cell proliferation was measured in real time by using a label-free, non-invasive cellular confluence assay by IncuCyte Live-Cell Imaging Systems (Essen Bioscience, Ann Arbor, MI, USA). 3000 hepatocytes/well were seeded on a collagen-coated 96-well plate, placed in the incubator maintained at 37°C with 5% $CO_2$ supply. The IncuCyte system scanned the plate and collected live cell images every 2 hr until the end of each experiment. The cell confluence was calculated using IncuCyte software and the cell proliferation is expressed as the percentage of confluence.

## Western blot analyses

Mouse liver tissues or cultured cells were homogenized in RIPA lysis buffer containing complete protease inhibitor cocktail and phosphatase inhibitor cocktail (Roche), resolved by SDS-PAGE, and transferred onto a PVDF membrane. The membranes were immunoblotted with antibodies against ERK (4695, Cell Signaling), Phospho-ERK (4370, Cell Signaling), JNK (9252, Cell Signaling), Phospho-JNK (4668, Cell Signaling), Phospho-cJun (3270, Cell Signaling), p38 (8690, Cell Signaling), Phospho-p38 (9215, Cell Signaling), AKT (9272, Cell Signaling), Phospho-AKT (4060, Cell Signaling), β-catenin (8480, Cell Signaling), Phospho-β-catenin (9565, Cell Signaling), Phospho-STAT3 (9145, Cell Signaling), Phospho-NF-κB p65 (3033, Cell Signaling), cMyc ( 5605, Cell signaling), Cyclin d1 (2978, Cell signaling), Anti-FlagM2 (F1804, Sigma-Aldrich), NLRP12 (Aviva Systems Biology, #OAAB04256), and β-actin (A2228, Sigma). Finally, immunoreactive proteins were detected using ECL super signal west femto substrate reagent (Thermo Scientific).

## Sandwich ELISA for quantitative measurement of cell signaling pathways

The activation of p65, ERK, JNK, p38, and STAT3 in the HCC of WT and *Nlrp12*$^{-/-}$ mice were measured by Pathscan Inflammation Multi-target Sandwich ELISA kit (Cell Signaling technology; #7276) and Pathscan phosphor-p44/42 MAPK Sandwich ELISA kit (Cell Signaling technology; #7177C) according to manufacturer's protocol. Briefly, HCC tissue lysates at a concentration of 0.5 mg/ml were added into antibody pre-coated ELISA plates and incubated overnight at 4°C. After washing, plates were incubated with HRP-conjugated detection antibodies. HRP substrate TMB was used to develop color and the absorbance was measured at 450 nm.

## Real-time PCR

Liver tissues were preserved in RNA later (Invitrogen). Total RNA was extracted using TRIzol (Invitrogen) according to the manufacturer's instructions. Isolated RNA was reverse transcribed into cDNA using iScript (Bio-Rad). Real-time PCR was performed using iTaq Universal SYBR Green Supermix (Bio-Rad). Expression data were normalized to GAPDH as described earlier (*Hu et al., 2015*). Primers used for real-time PCR are listed in *Supplementary file 1*-Table 1 and 2.

## Transient transfection and preparation of stable cell lines

The human NLRP12 cDNA was prepared from Jurkat cell cDNA library by PCR using oligonucleotide primers acttAAGCTTatgctacgaaccgcaggcagg and gtcgGATATCtgcagccaatgtccaaataa. The human NLRP12 cDNA was cloned into flag-tagged pcDNA4/TO vector, a CMV expression vector, at HindIII and EcoRV sites. The mouse NLRP12 cDNA was obtained from mouse macrophage by primers gtcgGATATCtgcagccaatgtccaaataag and tcgaGCGGCCGCccacacccaatatccaggtacgg. Then mouse NLRP12 cDNA was cloned into Flag-tagged pcDNA4/TO vector at KpnI and NotI sites. As a control, *GFP* was cloned into the pcDNA4/TO:Flag vector at BamHI and NotI sites. Mouse primary hepatocytes and HepG2 hepatocellular carcinoma cells were transiently transfected with mouse and human *NLRP12* construct respectively or *GFP* construct using lipofectamine 3000 reagent (Invitrogen). To make stable HepG2 cells expressing human *NLRP12* or *GFP*, the cells were transfected with *NLRP12* or *GFP* construct using lipofectamine 3000 reagent (Invitrogen), and were selected on Zeocin (100 μg/ml) and confirmed by observing GFP under fluorescence microscope and western blot analysis of Flag. HepG2 cells were cultured in DMEM supplemented with 10% FBS and 1% penicillin and streptomycin in a 5% $CO_2$ incubator at 37°C. The stable HepG2 cells expressing NLRP12 or GFP were maintained in Zeocin (100 μg/ml) containing DMEM supplemented with 10% FBS and 1% pen/strep.

## Knock down of NLRP12 in HepG2 cells

The human liver cancer cell line HepG2 was cultured in Dulbecco's Modified Eagle's medium (high glucose, Sigma) supplemented with 10% FBS and 1% penicillin and streptomycin (Sigma) and maintained in a 5% $CO_2$ incubator at 37°C. At 50–60% confluency, cells were transfected with either NLRP12 sgRNA CRISPR/Cas9 (K1434706, abm) or Scrambled sgRNA CRISPR/Cas9 (K010, abm) plasmids using Lipofectamine 3000 reagent (Invitrogen) according to manufacturer's instructions. 48 hr post-transfection, CRISPR/Cas9 plasmid-transfected HepG2 cells were selected using media containing 2 μg/ml puromycin (A1113803, Gibco). Knock down of NLRP12 in CRISPR/Cas9-transfected cells was confirmed by Western blot analysis. The NLRP12 or scrambled sgRNA transfected HepG2 cells were seeded in 12-well plate, incubated for overnight and stimulated with LPS (1 μg/ml). The activation of signaling pathways was measured by Western blotting.

## TUNEL assay

TUNEL staining was performed by TUNEL assay kit (In Situ Cell Death Detection Kit- Fluorescein, Cat # 11684795910, Roche) according to manufacturer's instruction. In brief, primary hepatocytes grown on cover slip were fixed with 4% paraformaldehyde. After permeabilization with 0.1% Triton X-100 in 0.1% sodium citrate, cells were incubated with TUNEL antibody. For liver tissues, 4% paraformaldehyde-fixed and paraffin-embedded tissue sections were de-paraffinized and hydrated through decreasing concentrations of ethanol. Antigen retrieval was done in 10 mM sodium citrate solution (pH 6.0) for 20 min at 95°C followed by treatment with TUNEL reagents. Nuclei of cells were counterstained with the DAPI reagent. Images were taken using Zeiss fluorescence microscope.

## Sequencing of 16S rRNA gene amplicons and analysis

Fecal samples were collected from 10-month-old WT and *Nlrp12*$^{-/-}$ mice. Using the Fecal DNA isolation kit (Qiagen, USA), total genomic DNA were extracted from fecal pellets. The concentration and purity of DNA (A260/280) were estimated spectrophotometrically. The quality of DNA was assessed by agarose gel electrophoresis. Bacterial primers 341F 5'-CCTACGGGAGGCAGCAG-3') and 806R (5'-GGACTACHVGGGTWTCTAAT-3') targeting the V3-V4 hyper-variable region of 16S rRNA gene was used for PCR amplification. In library preparation, sample-specific barcode sequences were incorporated to the primers, and then barcoded 16S rRNA gene amplicons were pooled at equimolar concentration prior to sequencing using MiSeq platform (Illumina, Inc, San Diego, California). The

raw sequence reads were demultiplexed according to the sample-specific barcodes, which followed by primer trimming and quality filtering, were assigned to operational taxonomic units (OTUs). The quality filtering steps include the removal of sequences with anonymous bases, chimera sequences using vsearch method (*Rognes et al., 2016*). In addition, the phred quality cutoff was set to 30 to maintain high quality sequences. We employed uclust method (*Edgar, 2010*) to cluster the quality-filtered sequence reads at a minimum of 97% sequence similarity using open reference OTU-picking approach, and assigned taxonomy to each sequence of the representative set using the GreenGenes database (*DeSantis et al., 2006*). Following the removal of the singleton OTUs, the sequence reads, which ranged from 12,194 to 26,028, were used for estimating the relative abundances of the bacterial taxa at phylum and family levels. All the upstream and downstream analyses of Illumina sequences were carried out in the QIIME 1.8.0 environment (*Caporaso et al., 2010*).

## Statistical analysis

Data for all in vivo study are presented as means ± SEM. In vitro experimental data are presented as means ± SD. Statistical significance was determined by two-tailed unpaired Student's t-test, and $p < 0.05$ was considered statistically significant. To compare the relative frequencies of bacterial taxa between control and treated groups, the nonparametric t-test was performed. GraphPad Prism eight software was used for statistical analyses.

## 16S rRNA raw sequence data accessibility

The 16S rRNA gene sequence data have been deposited to Sequence Read Archive (SRA) of National Center for Biotechnology Information (SRA accession SRP175050). Detailed information on the individual sample can be accessed at https://www.ncbi.nlm.nih.gov/Traces/study/?acc= SRP175050 with NCBI BioProject accession PRJNA512540.

## Acknowledgements

We would like to thank the UT Southwestern Animal Resource Center (ARC) for maintenance and care of our mouse colony. We are thankful to Millennium Pharmaceuticals and Dr. Thirumala-devi Kanneganti at St. Jude Children's Research Center for sharing *Nlrp12*[-/-] mice. We also thank Dr. James S Malter at the Department of Pathology, UT Southwestern Medical Center, for critically reviewing the manuscript. This work was supported by Cancer Prevention and Research Institute of Texas (CPRIT) Individual Investigator Award (RP160169), and UT Southwestern funding given to Hasan Zaki. Hao Zhu was supported by NCI (R01CA190525) and CPRIT (RP180268)

## Additional information

### Funding

| Funder | Grant reference number | Author |
|---|---|---|
| Cancer Prevention and Research Institute of Texas | RP160169 | Hasan Zaki |
| UT Southwestern Medical Center | | Hasan Zaki |

The funders had no role in study design, data collection and interpretation, or the decision to submit the work for publication.

### Author contributions

SM Nashir Udden, Data curation, Formal analysis, Investigation, Methodology, Writing—original draft; Youn-Tae Kwak, Investigation, Methodology; Victoria Godfrey, Data curation, Investigation; Md Abdul Wadud Khan, Formal analysis, Analyzed 16S rRNA sequencing; Shahanshah Khan, Investigation; Nicolas Loof, Resources, Methodology; Lan Peng, Investigation, Performed histopathological scorings; Hao Zhu, Resources, Writing—review and editing; Hasan Zaki, Conceptualization, Resources, Data curation, Formal analysis, Supervision, Funding acquisition, Validation, Investigation, Visualization, Methodology, Writing—original draft, Project administration, Writing—review and editing

## Author ORCIDs
Hao Zhu https://orcid.org/0000-0002-8417-9698
Hasan Zaki http://orcid.org/0000-0001-9002-5399

## Ethics
Animal experimentation: This study was performed under the protocol #2016-101683 which was approved by the Institutional Animal Care and Use Committee (IACUC). All animal experiments were conducted in accordance with the IACUC guidelines and the National Institutes of Health Guide for the Care and Use of Laboratory Animals.

## Decision letter and Author response
Decision letter https://doi.org/10.7554/eLife.40396.037
Author response https://doi.org/10.7554/eLife.40396.038

# Additional files

### Supplementary files
• Supplementary file 1. List of primers of mouse and human genes. All primers used in this study to measure the expression of mouse genes (Supplemental Table 1) and of human genes (Supplemental Table 2) by real-time qPCR analysis.
DOI: https://doi.org/10.7554/eLife.40396.030
• Transparent reporting form
DOI: https://doi.org/10.7554/eLife.40396.031

### Data availability
All data generated or analyzed during this study are included in the manuscript and supplemental materials. 16S rRNA gene sequencing of gut bacteria was analyzed and the raw data were submitted to NCBI. Data source and accession numbers were included in the manuscript.

The following datasets were generated:

| Author(s) | Year | Dataset title | Dataset URL | Database and Identifier |
|---|---|---|---|---|
| Udden SMN, Kwak YT, Godfrey V, Khan MAW, Loof N, Peng L, Zhu H, Zaki H | 2019 | NLRP12 WT and KO mice fecal samples Raw 16S rRNA sequence reads | https://www.ncbi.nlm.nih.gov/Traces/study/?acc=SRP175050 | NCBI SRP, SRP175050 |
| Udden SMN, Kwak YT, Godfrey V, Khan MAW, Khan S, Loof N, Peng L, Zhu H | 2019 | NLRP12 WT and KO mice fecal samples Raw 16S rRNA sequence reads | https://www.ncbi.nlm.nih.gov/bioproject/PRJNA512540 | NCBI BioProject, PRJNA512540 |

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
