## [Decision Letter]

Thank you for submitting your article "NLRP12 suppresses hepatocellular carcinoma via downregulation of cJun N-terminal kinase activation in the hepatocyte" for consideration by *eLife*. Your article has been reviewed by two peer reviewers, and the evaluation has been overseen by a Reviewing Editor and Tadatsugu Taniguchi as the Senior Editor. The reviewers have opted to remain anonymous.

The reviewers have discussed the reviews with one another and the Reviewing Editor has drafted this decision to help you prepare a revised submission.

Summary:

NLRP12 negatively regulates HCC pathogenesis and prevents HCC by maintaining a stable immunolandscape via downregulation of JNK-dependent inflammation and proliferation of hepatocytes.

Essential revisions:

1) More accurate quantitative methods are required to confirm the activation of JNK in NLRP12-inhibited HCC pathogenesis by excluding the involvement of the well-known NF-B and STAT3 pathways in this process.

2) The roles of NLRP12 in immune modulation have been ignored to some extent. The contribution of NLRP12-mediated modulation of immune response should be properly evaluated.

*Reviewer #1:*

In general the study provides a straight forward concept: Nlrp12 plays an important role in maintaining a stable immunolandscape through the regulation of JNK signaling to prevent HCC. However, the clinical patient data demonstrates that NLRP2 is more likely to be oncogenic in HCC than downregulated, in contrary to their model as there are three independent dataset that show a higher propensity for gene amplification in HCC tumors than deletions, which focus on how the KO of NLRP12 increases the tumor burden of chemically induced HCC. How do the authors reconcile that gene amplification also contributes to HCC (its potential role as oncogenic), while the main message is that the loss of NLRP2 is important for HCC (tumor suppressive)? While Figure 1B addresses that the down-regulation of NLRP12 is found in HCC, what is the likelihood of this data something that is important for HCC and not a specific event found in only in the UALCAN webportal? In addition, the tumors shown are only liver tumors, is this something specific to the liver or is are there nodules in other organs as well? Since DEN induces malignancy in other tissues as well. Is this likely to liver specific event as eluded by the author? Why would CCL_4_ treatment in DEN induced NLRP12 KO mice decrease the tumor size of DEN NLRP12 KO mice (Figure 1E vs. Figure 1J)? In addition, the mice seem to weigh less as well. These data indicate that CCL_4_, further hepatic injury, almost decreases tumor burden in DEN mice, and making them sicker. This doesn't make sense to the main paper. Have the authors looked at NLRP12 KO mice + CCL_4_ alone? Do these NLRP12 KO mice + CCL_4_ also have a higher mortality rate than NLRP12 KO mice +DEN alone? Does NLRP12 KO Mice +CCL_4_ also have a higher tumor burden then WT? These are more appropriate questions to ask as a secondary model. Survival is important indicator of tumor burden, does the NLRP12 KO+DEN mice get sicker and die sooner than others after the 10 months? These data would suggest that the viability of the tumor type is specific to NLRP12 KO and not the DEN itself.

*Reviewer #2:*

In the manuscript by Udden and colleagues, the authors report that NLRP12 negatively regulates HCC pathogenesis via downregulation of JNK-dependent inflammation and proliferation of hepatocytes. The role of NLRP12 as a negative regulator in multiple immune cells has been widely explored. However, the role and significance of NLRP12 in HCC pathogenesis have not been completely elucidated. The current study revealed a novel role of NLRP12 in hepatocyte during HCC pathogenesis. Moreover, the authors proposed that NLRP12-mediated JNK inhibition, but not NF-κB and ERK signaling, was critical for HCC prevention. Therefore, the study is of significance and novelty in the field of both immunology and oncology. Generally, the study is well designed and presented. Most of the data are convincing and impressive. However, there are several concerns may need to be further clarified.

1) The authors proposed a novel role of NLRP12 in hepatocytes during HCC pathogenesis while they also linked the effects to intestinal microbiota. However, since the model used is the NLRP12 knockout mice, the effects of NLRP12 in HCC pathogenesis definitely involved the immune microenvironment. While the data in Figures 1-3 are convincing, the roles of NLRP12 in immune modulation have been ignored to some extent. This is a major concern. The contribution of NLRP12-mediated modulation of immune response has not been properly evaluated. Additionally, whether NLRP12 KO affected the microbiota (leading to enhanced inflammation of liver)? Hepatocyte-specific knockout model may improve the convincingness of the study.

2) In experiments regarding responsiveness of hepatocytes to PAMPs, LPS stimuli were used as standard treatment. Although I understand LPS derived from microbiota may be the major PAMPs, the other major or well-known endogenous PAMPs generated during HCC pathogenesis may also be taken into consideration.

3) In Figure 4, the authors highlighted the JNK activation in NLRP12 KO tumors and cells. However, these data cannot convincingly exclude the activation of NF-κB and ERK pathway and the activation of STAT3 during HCC pathogenesis, as demonstrated in previous studies. Could the authors more quantitatively analyze the activation status of these related signaling mediators (WB is not sufficient as exclusion experiments)?

[Editors' note: further revisions were requested prior to acceptance, as described below.]

Thank you for resubmitting your work entitled "NLRP12 suppresses hepatocellular carcinoma via downregulation of cJun N-terminal kinase activation in the hepatocyte" for further consideration at *eLife*. Your revised article has been favorably evaluated by Tadatsugu Taniguchi (Senior Editor), a Reviewing Editor, and two reviewers.

The manuscript has been improved but there are some remaining issues that need to be addressed before acceptance, as outlined below:

It would be nice if the authors could provide data that NLRP12 is downregulated in hepatocytes of DEN- or DEN plus CCL_4_-induced HCC in mice or that NLRP12 protein level is downregulated in human HCC tissues. Please discuss the weaknesses of this study: the lack of hepatocyte-specific Nlrp12 KO models and the fact that the conclusion on the contributions of inflammatory cell populations and cytokines to HCC pathogenesis in this model would need to be supported by more data in the future.

*Reviewer #1:*

The authors investigated the role of Nlrp12 as a protective protein against carcinogen-induced HCC via animal models. The authors show that in DEN Nlrp12 KO mice had high levels of proto-oncogenes and activation of JNK signaling. Inhibition of JNK signaling reduced proliferation and hepatocyte inflammatory responses.

The authors indicate in the rebuttal that the both genetic mutations and gene amplifications can occur and play contradicting roles in cancer. I agree. But the authors suggest that NRPL12 knockout enhances HCC progression. The concern here is the interpretation of what these genomic data means. A low frequency of alterations (<3%) in NRPL12 indicate that in humans, NRPL12 genomic alterations are rare and that NRPL12 has the potential to play both roles. In addition, the authors work shows that NRPL12 KO enhances HCC progression, therefore its likely not important for tumor initiation (as the NRPL12 KO alone doesn't produce much tumors). In addition, the author did not reconcile that in humans, NRPL12, with its low frequency in changes and a 1% vs. 2.5% in TCGA, but a 1% and 1% in AMC, or mutated less 2% in Inserm, that NRPL12 potentially could also be oncogenic and potentially play a minimal role in HCC progression? What is the likelihood that all other gene mutations with similar alterations in human samples are also functionally important, if not more important? While I appreciate that it's not easy to do a holistic approach, this data was not well explained nor was it reconciled in the manuscript or carefully and thoughtfully rebutted.

In addition, the RNASeq data (using a different data platform (UALCAN instead of cbioportal, which collects and processes TCGA differently, also is an issue as it may skew data). The authors indicate that because they used UALCAN, a platform for TCGA analysis, for RNASeq values data, the graph 1B is considered unbiased? I'm not sure what that means. This figure shows that in the TCGA data, there is a difference between tumor and non-tumor (not the same as normal livers, since TCGA patients are more likely to be HBV, HCV, NAFLD and fatty liver disease as underlying liver disease) with unequal numbers of samples. What is the likelihood of this difference being specific to this dataset (could be background noise due to the low number of non-tumor samples) as opposed to be a robust HCC specific observation? To answer this question, the authors have to check other datasets that are available to confirm that these observations are stable (not specific to this cohort, not specific to the low numbers of non-tumors). The authors did so in 1A, but did not in 1B.

In response to the authors comment about DEN-induced mice not producing tumors in other organs, DEN is also widely used as a carcinogen in other animal models for other cancer types. Where are the data to demonstrate this model doesn't? I don't agree that because its well-used in the liver field, the authors shouldn't show controls. At least at the timepoint in which the authors sacrificed the mice.

*Reviewer #2:*

The authors have performed additional experiments to address my concerns, such as comparison of microbiota, ELISA confirmation of JNK activation, and clarification of inflammatory factors in HCC pathogenesis. More importantly, the authors have clearly described the limitations and provided possibilities for the current data. So generally I agree that the authors have made essential revisions for the manuscript. Considering the importance and novelty of the current study, I recommend to accept the paper for publication.

---

## [Author Response]

Essential revisions:1) More accurate quantitative methods are required to confirm the activation of JNK in NLRP12-inhibited HCC pathogenesis by excluding the involvement of the well-known NF-B and STAT3 pathways in this process.

To understand what signaling pathways contribute to increased HCC pathogenesis in *Nlrp12^-/-^* mice, we measured the activation status of different signaling pathways including NF-κB (p65), ERK, JNK, p38, and STAT3 in tumor lysates from wild-type and *Nlrp12^-/-^* mice by Western blotting (Figure 4D and 4E). Our data show that all these pathways are activated in both WT and *Nlrp12^-/-^* HCC. However, the intensity of P-JNK was significantly higher in *Nlrp12^-/-^*HCC compared to that of WT (Figure 4D and 4E). We would like to clarify that we are not implying that JNK is the only pathway activated in *Nlrp12^-/-^* HCC tissues and other pathways do not contribute to HCC. While p65, ERK, p38, and STAT3 pathways are also activated, there was no statistically significant difference in their levels between WT and *Nlrp12^-/-^*tumors (Figure 4E), implicating a dominant role of P-JNK in higher HCC pathogenesis in *Nlrp12^-/-^* mice.

Western blot is a gold standard method for measuring the activation of cell signaling pathways. However, as the reviewer insisted on measuring the activation of these pathways in a different quantitative approach, we used an ELISA-based method (Pathscan Inflammation Multi-Target ELISA; Cell Signaling Technology) (Figure 4F). The new data showing higher P-JNK in *Nlrp12^-/-^* HCC lysates compared to WT is consistent with our Western blot analysis. There was no significant difference in the activation of other pathways (Figure 4F). Therefore, this quantitative analysis further strengthens our observation that NLRP12 down-regulates JNK activation in the liver during HCC development.

2) The roles of NLRP12 in immune modulation have been ignored to some extent. The contribution of NLRP12-mediated modulation of immune response should be properly evaluated.

To understand the role of NLRP12 in immunomodulation, we measured the expression of several proinflammatory cytokines and chemokines in HCC tissues by real-time qPCR (Figure 2C, Figure 2—figure supplement 1D), and quantify different immune cell population including macrophages, dendritic cells, T cells, and neutrophils by flow cytometry (Figure 2D and 2E). We observed that IL-6, TNFα, Cxcl11 (KC), Cxcl2 (MIP2), and Ccl2 (MCP1) are significantly highly expressed in the tumors of *Nlrp12^-/-^* mice. In addition to these, we measured several others cytokines and chemokines but no difference was observed in their levels between the groups. In the interest of space, we didn’t include those non-significant data. The flow cytometric analysis of tumor infiltrated immune cells show a higher number of macrophages and dendritic cells in *Nlrp12^-/-^* tumors, while there was no major difference in T cells and neutrophils (Figure 2D and 2E). Consistently, we could not see any difference in IFNγ, IL-17 and IL-4 (Figure 2—figure supplement 1F), suggesting that NLRP12 doesn’t regulate T cell responses in HCC. Since we didn’t observe any difference in T cell number or T-cell dependent cytokines between wild-type and *Nlrp12^-/-^* HCC, we didn’t further investigate the role of NLRP12 in T cell responses in the liver. Instead, we focused on understanding why *Nlrp12^-/-^* HCC exhibited higher levels of proinflammatory cytokines and chemokines. To this end, we analyzed signaling pathways and observed that Nlrp12 deficiency leads to hyper activation of JNK in the HCC. We further clarified what cell types regulate JNK activation in Nlrp12-dependent manner by isolating hepatocytes, Kupffer cells, and hepatic stellate cells. Our data suggest that Nlrp12 regulates JNK activation in the hepatocyte but not in macrophages and stellate cells (Figure 4J). All these data led us to focus on the hepatocyte-specific function of NLRP12 which we described in Figure 6 and Figure 7. With all these data, we believe that we investigate the role of NLRP12 in immune modulation to an extent what is necessary to explain the underlying mechanism of higher HCC susceptibility in *Nlrp12^-/-^* mice.

As the second reviewer stressed to further analyze the role of NLRP12 in the regulation of immune responses, we performed additional experiments and included the new data in the revised manuscript. At first, we measured the key cytokines IL-6, TNFα, and KC in the lysate of WT and *Nlrp12^-/-^* tumors by ELISA. We observed that, consistent to real-time qPCR analysis, these proinflammatory cytokines were significantly highly expressed in *Nlrp12^-/-^*HCC (Figure 2—figure supplement 1E). Next, we addressed whether NLRP12 deficiency affects immune responses in the liver at homeostasis. To this end, we measured cytokines and chemokines in age and sex-matched healthy WT and *Nlrp12^-/-^* mouse livers. The expression levels of IL-6, TNFα, CXCL1, CXCL2, and CCL2 were seen comparable between WT and *Nlrp12^-/-^* livers (Figure 5—figure supplement 1C). We also measured the number of different immune cell population in the liver of age-matched healthy WT and *Nlrp12^-/-^* mice by flow cytometry and observed no difference in the number of macrophages, dendritic, T cells, and neutrophils between the two groups (Figure 5—figure supplement 1D). This data is supported by real-time qPCR analysis of F4/80 in the healthy liver, showing no major difference between the groups (Figure 2—figure supplement 1G). Finally, we measured the activation status of different signaling pathways in healthy livers. Interestingly, no difference in the activation of JNK, as well as other signaling pathways, was observed between healthy WT and *Nlrp12^-/-^* livers (Figure 5—figure supplement 1E). All these data suggest that Nlrp12 deficiency doesn’t induce any immune dysregulation in the liver without any pathological trigger. We have discussed these observations in the last paragraph of the subsection “NLRP12 attenuates PAMPs-mediated hepatic inflammation and oncogenesis”.

In summary, our data suggest that NLRP12-mediated regulation of JNK in the hepatocyte play the key role in the suppression of HCC. Previously, we and others have shown that NLRP12 regulates inflammatory responses in myeloid and T cells in different disease contexts (Zaki et al., 2011; Lukens et al., 2015). It is possible that NLRP12 regulates different signaling pathways in cell type and tissue-specific manner. However, we do not completely exclude the involvement NLRP12-dependent regulation of myeloid and T cell-mediated immune responses in HCC pathogenesis. To further characterize NLRP12-dependent regulation of immune responses during HCC, we need to conditionally knockout NLRP12 in different immune cells. Due to resource limitation, we are not able to extend our investigation to that extent in this current study, but our future study will focus on this aspect.

Reviewer #1:In general the study provides a straight forward concept: Nlrp12 plays an important role in maintaining a stable immunolandscape through the regulation of JNK signaling to prevent HCC. However, the clinical patient data demonstrates that NLRP2 is more likely to be oncogenic in HCC than downregulated, in contrary to their model as there are three independent dataset that show a higher propensity for gene amplification in HCC tumors than deletions, which focus on how the KO of NLRP12 increases the tumor burden of chemically induced HCC. How do the authors reconcile that gene amplification also contributes to HCC (its potential role as oncogenic), while the main message is that the loss of NLRP2 is important for HCC (tumor suppressive)?

We thank the reviewer for the valuable comments. With due respect to the reviewer comments, we disagree with the reviewer’s opinion that NLRP12 is more likely to be oncogenic. The data presented in Figure 1A shows that mutation in *NLRP12* is seen in 1.5-2.5% HCC, while amplification is observed in less than 1% patients. It is not uncommon that both mutation and amplification of a particular gene are associated with carcinogenesis. While loss-of-function mutations of a particular gene are commonly associated with carcinogenesis, amplification of the same gene is also linked with cancer susceptibility. Nlrp12 amplification may reduce JNK activation to a point that dysregulates normal liver physiology and promotes tumorigenesis. Indeed, inhibition of JNK1 and JNK2 in hepatocytes was seen to promote HCC burden (Das et al., 2011), while *Jnk1^-/-^* mice were protected from HCC (Hui et al., 2008). Similarly, TCGA database analyses through cBioportal show that both mutation and amplification of JNK are linked with HCC. Thus, it is not surprising that both mutations and amplifications of NLRP12 are associated with increased HCC.

While Figure 1B addresses that the down-regulation of NLRP12 is found in HCC, what is the likelihood of this data something that is important for HCC and not a specific event found in only in the UALCAN webportal?

UALCAN is a platform for analysis of publicly available RNAseq database of cancer. The graph presented here (Figure 1B) was based on TCGA data, which is a great resource of cancer genomics and commonly used by the research community. Therefore, the analysis shown in Figure 1B should be considered unbiased.

In addition, the tumors shown are only liver tumors, is this something specific to the liver or is are there nodules in other organs as well? Since DEN induces malignancy in other tissues as well. Is this likely to liver specific event as eluded by the author?

DEN-induced HCC is widely used model for HCC study. To our knowledge, the dose of DEN we used does not induce tumor in other organs. Indeed, we didn’t observe any tumor development in other organs.

Why would CCL_4_ treatment in DEN induced NLRP12 KO mice decrease the tumor size of DEN NLRP12 KO mice (Figure 1E vs. Figure 1J)? In addition, the mice seem to weigh less as well. These data indicate that CCL_4_, further hepatic injury, almost decreases tumor burden in DEN mice, and making them sicker. This doesn't make sense to the main paper.

The differences in tumor size and mouse body between DEN plus CCl_4_ vs DEN alone model is due to the difference in time points when HCC was evaluated. As outlined in these two models in Figure 1C and 1H, the endpoints of the DEN model and DEN +CCl_4_ model were 10 and 6 months after birth respectively. No visible tumor lesion was observed in mice treated with DEN alone until 7 months post DEN administration (data not shown). Therefore, had we compared livers of DEN-treated mice at 6 months, we would not see any tumors in the earlier group while DEN+CCl_4_-treated mice developed an appreciable number of visible tumors. Similarly, if we would have allowed DEN+CCl_4_-treated mice to live until 10 months, there would have a higher number and much bigger tumors in the liver. Even some mice may not survive up to that time point due to severe HCC related complication. Therefore, the data presented here don’t imply that CCl_4_ suppresses DEN-induced HCC, rather it accelerates HCC development. For the same reason, the reviewer might be confused with the data representing liver to body weight ratios (Figure 1E and 1J). Since DEN-treated mice were aged (10 months) compared to DEN+CCl_4_-treated mice (6 months), the body weight of the former group was higher. Furthermore, mice received CCl_4_ for 2 months starting at 2.5 months after their birth. CCl_4_ is a hepatotoxin which induces liver injury and affects liver function. Thus DEN+CCl_4_-treated mice gained body weight at a slower rate than DEN-treated mice. However, both groups of mice remained apparently healthy until the endpoint and there was no incidence of death during the experiment. Finally, we would like to mention that both of these two models are commonly used in HCC studies. The number and size of tumor lobes depend on the dose of CCl_4_and the duration of the experiment. Thus, although the tumor size and other morphometric features of HCC obtained from two different models are not comparable, the conclusion that *Nlrp12^-/-^* mice are more susceptible to HCC compared to WT mice is same.

*Have the authors looked at NLRP12 KO mice +* CCl_4_
*alone? Do these NLRP12 KO mice +* CCl_4_
*also have a higher mortality rate than NLRP12 KO mice +DEN alone? Does NLRP12 KO Mice +*CCl_4_
*also have a higher tumor burden then WT? These are more appropriate questions to ask as a secondary model. Survival is important indicator of tumor burden, does the NLRP12 KO+DEN mice get sicker and die sooner than others after the 10 months? These data would suggest that the viability of the tumor type is specific to NLRP12 KO and not the DEN itself.*

Thanks for asking some intriguing questions. Without DEN, CCl_4_ alone doesn’t induce any tumor and no DEN or DEN+CCl_4_-injected mouse died during the experiment. The role of NLRP12 on CCl_4_-induced liver fibrosis is an important concern, but out of the scope of this current study. A role for NLRP12 in liver fibrosis should be investigated in depth in a separate study using appropriate animal models (we discussed in the last paragraph of the subsection “NLRP12 regulates hepatocyte proliferation via JNK”). We agree that it would be interesting to see whether DEN-treated *Nlrp12^-/-^*mice die faster than WT mice. However, our IACUC protocol does not permit to leave the experimental mice until they die. With the help of published literature and our own preliminary experiments, we optimized treatment doses and experimental scheme which would allow us to evaluate HCC pathogenesis without reaching to a condition when mice would become moribund. We are required by the IACUC to terminate the study before mice suffer from pain and distress.

Reviewer #2:[…] 1) The authors proposed a novel role of NLRP12 in hepatocytes during HCC pathogenesis while they also linked the effects to intestinal microbiota. However, since the model used is the NLRP12 knockout mice, the effects of NLRP12 in HCC pathogenesis definitely involved the immune microenvironment. While the data in Figures 1-3 are convincing, the roles of NLRP12 in immune modulation have been ignored to some extent. This is a major concern. The contribution of NLRP12-mediated modulation of immune response has not been properly evaluated. Additionally, whether NLRP12 KO affected the microbiota (leading to enhanced inflammation of liver)? Hepatocyte-specific knockout model may improve the convincingness of the study.

Please see our reply in response to editorial comment #2 regarding the concern on the role of NLRP12 in the modulation of immune response.

To address the reviewer’s concern regarding microbiota, we measured the microbiota composition of healthy wild-type and *Nlrp12^-/-^* mice by 16S rRNA gene sequencing. We observed that the composition of certain bacterial species is different between WT and *Nlrp12^-/-^* mice. However, such a difference in microbiota composition didn’t impact on immune homeostasis in the liver as inflammatory responses in the livers of the healthy WT and *Nlrp12^-/-^* mice are comparable (Figure 5—figure supplement 1C-E). We have discussed the new data in the last paragraph of the subsection “NLRP12 attenuates PAMPs-mediated hepatic inflammation and oncogenesis”.

2) In experiments regarding responsiveness of hepatocytes to PAMPs, LPS stimuli were used as standard treatment. Although I understand LPS derived from microbiota may be the major PAMPs, the other major or well-known endogenous PAMPs generated during HCC pathogenesis may also be taken into consideration.

We have shown that in addition to LPS, other PAMP_S_ such as peptidoglycan (PGN) and cytokine TNFa also regulate *JNK* in an Nlrp12-dependent manner (Figure6—figure supplement 1B and 1C). Several molecules and danger signals produced in the tumor microenvironment may also activate *JNK*. However, examining the role of NLRP12 in activating *JNK* in response to all of these endogenous ligands and implicating those data in our current study is out of context.

3) In Figure 4, the authors highlighted the JNK activation in NLRP12 KO tumors and cells. However, these data cannot convincingly exclude the activation of NF-κB and ERK pathway and the activation of STAT3 during HCC pathogenesis, as demonstrated in previous studies. Could the authors more quantitatively analyze the activation status of these related signaling mediators (WB is not sufficient as exclusion experiments)?

Please see our reply to the editorial comment #1.

[Editors' note: further revisions were requested prior to acceptance, as described below.]

The manuscript has been improved but there are some remaining issues that need to be addressed before acceptance, as outlined below:

*It would be nice if the authors could provide data that NLRP12 is downregulated in hepatocytes of DEN- or DEN plus* CCL_4_*-induced HCC in mice or that NLRP12 protein level is downregulated in human HCC tissues. Please discuss the weaknesses of this study: the lack of hepatocyte-specific Nlrp12 KO models and the fact that the conclusion on the contributions of inflammatory cell populations and cytokines to HCC pathogenesis in this model would need to be supported by more data in the future.*

We thank the editors and reviewers to review our revised manuscript favorably. As suggested by the editors, we measured Nlrp12 expression in healthy and HCC livers and presented the data in the revised manuscript (Figure 1C). The new data show that Nlrp12 is downregulated in HCC tissue compared to healthy livers. This data supports the RNAseq data showing that the expression of *NLRP12* is significantly reduced in human HCC (Figure 1B).

We have discussed the weakness of this study as the reviewer and editors pointed out in the revised manuscript. Please see the fourth paragraph of the Discussion for relevant changes.

Reviewer #1:[…] The authors indicate in the rebuttal that the both genetic mutations and gene amplifications can occur and play contradicting roles in cancer. I agree. But the authors suggest that NRPL12 knockout enhances HCC progression. The concern here is the interpretation of what these genomic data means. A low frequency of alterations (<3%) in NRPL12 indicate that in humans, NRPL12 genomic alterations are rare and that NRPL12 has the potential to play both roles. In addition, the authors work shows that NRPL12 KO enhances HCC progression, therefore its likely not important for tumor initiation (as the NRPL12 KO alone doesn't produce much tumors). In addition, the author did not reconcile that in humans, NRPL12, with its low frequency in changes and a 1% vs. 2.5% in TCGA, but a 1% and 1% in AMC, or mutated less 2% in Inserm, that NRPL12 potentially could also be oncogenic and potentially play a minimal role in HCC progression? What is the likelihood that all other gene mutations with similar alterations in human samples are also functionally important, if not more important? While I appreciate that it's not easy to do a holistic approach, this data was not well explained nor was it reconciled in the manuscript or carefully and thoughtfully rebutted.

We thank the reviewers for critically reviewing our manuscript, which helped improve our manuscript and better explain the role of NLRP12 in HCC. We agree with the reviewer that NLRP12-deficiency alone doesn’t initiate HCC. During tumor promotion, NLRP12 acts as a tumor suppressor, at least in our mouse model. Therefore Nlrp12-deficient mice develop a higher number and faster progression of the tumor. We don’t consider NLRP12 a potential oncogene or tumor suppressor, rather it is a regulator of signaling pathways, such as JNK, which are involved in HCC pathogenesis. Similar to NLRP12, many other innate immune molecules are involved in the regulation of HCC. Previous studies described the role of JNK, IKKb, NEMO, RIG-I, TLR4, etc. in HCC pathogenesis (Dapito et al., 2012; Hou et al., 2014; Hui et al., 2008; Luedde et al., 2007; Maeda et al., 2005). None of these innate molecules are seen to be altered at a higher frequency in human HCC (please see Author response image 1), but their roles in the protection against HCC are appreciated. In fact, HCC is a genetically heterogeneous disease. While several genes such as *TP53, CTCNB1, WNT, ARID1A, TERT*, etc. are frequently mutated in HCC, genetic alterations in many other genes are associated with HCC and has been shown to promote the tumorigenesis process. Therefore, many genes with lower frequency mutations are considered critical regulators of HCC pathogenesis. As the reviewer may understand, comparison of NLRP12 with other related genes in HCC pathogenesis is out of scope of this study. However, we have added a discussion on this issue in light of the reviewer’s comments in the revised manuscript (please see Discussion, second paragraph). We hope that with these changes and discussion in this rebuttal letter, the reviewer will find that we have adequately addressed the reviewer’s concerns.

**Author response image 1. respfig1:** Analysis of gene alterations of different innate immune genes using cBioportal.

In addition, the RNASeq data (using a different data platform (UALCAN instead of cbioportal, which collects and processes TCGA differently, also is an issue as it may skew data). The authors indicate that because they used UALCAN, a platform for TCGA analysis, for RNASeq values data, the graph 1B is considered unbiased? I'm not sure what that means. This figure shows that in the TCGA data, there is a difference between tumor and non-tumor (not the same as normal livers, since TCGA patients are more likely to be HBV, HCV, NAFLD and fatty liver disease as underlying liver disease) with unequal numbers of samples. What is the likelihood of this difference being specific to this dataset (could be background noise due to the low number of non-tumor samples) as opposed to be a robust HCC specific observation? To answer this question, the authors have to check other datasets that are available to confirm that these observations are stable (not specific to this cohort, not specific to the low numbers of non-tumors). The authors did so in 1A, but did not in 1B.

The major focus of this study is to demonstrate a biological function of NLRP12 in inflammation and carcinogenesis in the liver using animal models of HCC. The reviewer may appreciate that we have demonstrated compelling data based on animal models and cell culture systems which support a critical role for NLRP12 in the suppression of HCC. We are very sincere about the robustness and reproducibility of our experimental data. Therefore, we performed every key experiment in multiple ways. For example, we showed the HCC phenotype in two different model – DEN and DEN plus CCl_4_. We measured the signaling pathways in whole liver, tumor tissues, isolated tumor cells, and primary hepatocytes. In addition, we overexpressed and knocked down NLRP12 in vitro and confirmed the function of NLRP12 in hepatocytes. Similarly, we established the role of NLRP12 in hepatocyte proliferation in multiple approaches including Ki67 staining of tumor tissue and hepatocyte culture, IncuCyte live imaging, and BrdU staining. Including mutation and expression analysis of NLRP12 in human HCC (Figure 1A and 1B) is a part of our effort to support the findings of our experimental study. However, genome sequencing and bioinformatics are not our expertise. We, therefore, relied on publicly available online platforms for genomics and transcriptomics analyses of NLRP12. We analyzed the frequency of alterations of *NLRP12* in human HCC based on available datasets in cBioportal. However, cBioportal is not a suitable platform for the analysis of expression profile of a gene of interest and comparison between normal vs tumor samples. We found that UALCAN web-portal, which uses only TCGA datasets, is a helpful platform in this regard. We are sorry that we are not able to provide RNAseq data analysis from other sources as the reviewer asked; UALCAN doesn’t have any resource/option to analyze other datasets. On the other hand, cBioportal doesn’t provide a comparative analysis of gene expression in tumor vs non-tumor samples. However, as the editor suggested, we now provide Nlrp12 expression profile in healthy and HCC mouse livers. Our data demonstrate that Nlrp12 expression is significantly reduced in liver tumors compared to healthy livers (Figure 1C). We hope that the reviewer will find this new data is supportive of human *NLRP12* expression data (Figure 1B).

We understand the importance of a large scale clinical data analysis to establish the role of a gene in a particular disease. However, we believe that our study will be used as a platform for future research focusing on genomics and transcriptomics analyses of NLRP12 in HCC and other liver disorders.

In response to the authors comment about DEN-induced mice not producing tumors in other organs, DEN is also widely used as a carcinogen in other animal models for other cancer types. Where are the data to demonstrate this model doesn't? I don't agree that because its well-used in the liver field, the authors shouldn't show controls. At least at the timepoint in which the authors sacrificed the mice.

As we explained previously, we opted to use DEN model because it is widely used for HCC study (Bakiri and Wagner, 2013). We acknowledge that as a DNA alkylating agent, DEN may have a potential for inducing tumor in other organs. However, DEN-induced carcinogenesis is dependent on dose, time of administration, and duration of the treatment (Verna et al., 1996). For example, a low dose of DEN administration in mice at 2 weeks of age induces tumor development in the liver because liver cells are rapidly proliferating at that age allowing DNA modification. DEN-induced HCC model was developed by other researchers about 5 decades ago (Rajewsky et al., 1966) and optimized over time to make it an ideal model for studying HCC. We followed HCC studies published by eminent scientists like Dr. Michael Karin and Dr. Robert Schwabe. In their published literature, we could not find any description of tumor development in other organs during DEN-induced HCC (Dapito et al., 2012; Maeda et al., 2005; Sakurai et al., 2008; Sakurai et al., 2006). In our own study, we also didn’t notice any apparent tumor development in other organs like kidney, liver, and gastrointestinal tract. We are showing some representative images here taken during the sacrifice of DEN-treated mice (please see Author response image 2). We are not excluding the possibility of tumor metastasis in other organs at an advanced stages of HCC. However, honestly speaking, we did not focus on this aspect in this current study. To test such a possibility, we need to examine different organs histopathologically. In the future study, we will investigate whether Nlrp12-deficiency promotes tumor metastasis.

**Author response image 2. respfig2:** Gross anatomy of DEN and DEN plus CCL_4_-treated mice.